# Gene-specific mechanisms direct glucocorticoid-receptor-driven repression of inflammatory response genes in macrophages

Maria A Sacta[1,2,3], Bowranigan Tharmalingam[2], Maddalena Coppo[2], David A Rollins[2,3], Dinesh K Deochand[2], Bradley Benjamin[2], Li Yu[4], Bin Zhang[5], Xiaoyu Hu[2,5], Rong Li[6], Yurii Chinenov[2], Inez Rogatsky[2,3]*

[1]Weill Cornell/ Rockefeller/ Sloan Kettering Tri-Institutional MD-PhD Program, New York, United States; [2]Hospital for Special Surgery Research Institute, The David Rosensweig Genomics Center, New York, United States; [3]Graduate Program in Immunology and Microbial Pathogenesis, Weill Cornell Graduate School of Medical Sciences, New York, United States; [4]Tsinghua-Peking Center for Life Sciences, Tsinghua University, Beijing, China; [5]Institute for Immunology and School of Medicine, Tsinghua University, Beijing, China; [6]Department of Molecular Medicine, The University of Texas Health Science Center at San Antonio, San Antonio, United States

*For correspondence: rogatskyi@hss.edu

Competing interests: The authors declare that no competing interests exist.

**Abstract** The glucocorticoid receptor (GR) potently represses macrophage-elicited inflammation, however, the underlying mechanisms remain obscure. Our genome-wide analysis in mouse macrophages reveals that pro-inflammatory paused genes, activated via global negative elongation factor (NELF) dissociation and RNA Polymerase (Pol)2 release from early elongation arrest, and non-paused genes, induced by de novo Pol2 recruitment, are equally susceptible to acute glucocorticoid repression. Moreover, in both cases the dominant mechanism involves rapid GR tethering to p65 at NF-kB-binding sites. Yet, specifically at paused genes, GR activation triggers widespread promoter accumulation of NELF, with myeloid cell-specific NELF deletion conferring glucocorticoid resistance. Conversely, at non-paused genes, GR attenuates the recruitment of p300 and histone acetylation, leading to a failure to assemble BRD4 and Mediator at promoters and enhancers, ultimately blocking Pol2 initiation. Thus, GR displays no preference for a specific pro-inflammatory gene class; however, it effects repression by targeting distinct temporal events and components of transcriptional machinery.
DOI: https://doi.org/10.7554/eLife.34864.001

## Introduction

Inflammation is an innate immune response to tissue injury or infection. It relies on macrophages, which recognize pathogen-associated molecular patterns and other 'danger' signals via their toll-like receptors (TLRs) (*Glass and Saijo, 2010*). This initiates a signaling cascade that leads to the activation and DNA binding of the effector transcription factors NF-kB and AP1 (*O'Neill et al., 2013*) which recruit coregulators, and, ultimately, the basal transcription machinery that together alter the chromatin state in the vicinity of many pro-inflammatory genes and enable their transcription (*Smale and Natoli, 2014*; *Glass and Natoli, 2015*). Acute transcriptional activation of pro-inflammatory genes is, therefore, critical for overriding the homeostatic set-point and producing a robust immune response that helps to resolve infection or tissue injury (*Kotas and Medzhitov, 2015*).

**eLife digest** Inflammation is one of the body's responses to fight infection and heal tissue damage. The response is controlled by hundreds of genes, which fall into two classes. In the first class, an injury or infection triggers the enzyme RNA Polymerase to bind to and transcribe the gene into long RNA strands, which are then translated into the proteins that play a role in the inflammation response. The second class has a more quick-fire response. RNA Polymerase binds to these genes even without an injury or infection to serve as a trigger. But most of the time the enzyme only transcribes the beginning of these genes. This is because it is inhibited by a so-called negative elongation factor, which acts like a brake. For this second class of genes, an infection or injury triggers the release of the negative elongation factor from the enzyme, and allows RNA Polymerase to transcribe the full RNA strand.

In excess, inflammation can be dangerous. The body's way of limiting or controlling inflammation is via steroid hormones called glucocorticoids. These bind to the glucocorticoid receptor, which acts to switch off the inflammatory genes. But exactly how the receptor does this has not been fully understood.

Sacta et al. investigated how the glucocorticoid receptor turns off these gene complexes. Experiments looking at white blood cells in mice found that the receptor can switch off both groups of inflammatory genes, but by a different mechanism for each class. Sacta et al. discovered that in the first gene class, the receptor blocks proteins that open up the DNA for RNA Polymerase, so it could not bind to the gene. In the second class, the receptor stops the release of the brake-like negative elongation factor from RNA Polymerase. As a result, the enzyme stalls at the beginning of the gene and fails to make a full-length transcript required to make the necessary protein.

Glucocorticoids are often used as drugs to treat chronic inflammation, but they can have debilitating side effects. Understanding how the glucocorticoid receptor switches off inflammatory genes could help to design drugs with fewer side effects to treat chronic inflammation, and diseases caused by specific inflammatory genes.

DOI: https://doi.org/10.7554/eLife.34864.002

Although the magnitude and dynamics of inflammation is affected at multiple levels, the temporal coordination of cytokine gene transcription by RNA Polymerase (Pol) 2 is a key mechanism that defines acute inflammatory response. The Pol 2 transcription cycle has been divided into three phases: initiation, elongation and termination. Initiation involves the recruitment of Pol 2 to the promoter, histone modifications and changes in histone occupancy. In addition, the C-terminal domain (CTD) of Pol 2, which contains multiple heptad repeats (YS2PTS5PS), is phosphorylated at S5, and Pol 2 synthesizes short (20–60 nt) RNA transcripts. During the elongation step, Pol 2 is further phosphorylated at S2 by the cyclin T1/CDK9 positive transcription elongation factor (P-TEFb) and synthesizes the full length RNA transcript, which is followed by the termination step and RNA transcript dissociation from the DNA (*Nechaev and Adelman, 2011*).

Although Pol 2 recruitment and initiation has been historically considered the rate-limiting step in signal-dependent transcription, numerous recent studies revealed that transcriptionally engaged Pol 2 often remains paused near promoters in the absence of activating signal and that entry into productive elongation is rate-limiting for activation of up to 40% of inducible genes (*Core et al., 2012*). The paused Pol 2 is in a complex with the 4-subunit negative elongation factor (NELF); NELF phosphorylation by P-TEFb leads to its release and Pol 2 entry into productive elongation (*Chiba et al., 2010*; *Nechaev and Adelman, 2011*). A subset of cytokine genes in macrophages is controlled at the level of Pol 2 pausing. Indeed, while for genes such as Il1a and Il1b, signal-dependent Pol 2 recruitment to their transcription start sites (TSS) and transcription initiation are rate-limiting, other genes, exemplified by Tnf, are occupied by Pol 2 even under resting conditions (*Adelman et al., 2009*; *Hargreaves et al., 2009*; *Gupte et al., 2013*). At Tnf, Pol 2 is S5-phosphorylated, bound by NELF and paused ~50 bp downstream of the TSS. Pause release following S2 and NELF phosphorylation by P-TEFb occurs in response to inflammatory signal.

Aside from Pol 2 occupancy, the chromatin state plays an integral part in the regulation of transcription (*Smale et al., 2014*). In particular, histone code 'writers' such as acetyltransferases (HATs)

GCN5 and p300 have been implicated in modifying H3K9/14 and H4K5/8/12 at inflammatory genes in macrophages following treatment with TLR4 ligands (*Hargreaves et al., 2009*; *Escoubet-Lozach et al., 2011*). Both HATs are recruited by the NF-kB subunit p65 to regulatory regions in a stimulus-dependent manner (*Hargreaves et al., 2009*; *Ghisletti et al., 2010*). Histone modifications are then bound by 'readers' such as BRD4, a protein containing two conserved N-terminal bromodomains (BD1 and BD2), which associates with most active promoters and some active enhancers, and has been proposed to couple the acetylation state at enhancers and promoters with Pol 2 elongation (*Lovén et al., 2013*; *Brown et al., 2014*). BRD4 occupancy correlates with acetylation marks at H4K5/8/12, H3K9/27 (*Lovén et al., 2013*; *Kanno et al., 2014*; *Nagarajan et al., 2014*) and with gene activation, whereas chemical inhibition of BRD4 binding abrogates the induction of a subset of genes (*Nicodeme et al., 2010*). Furthermore, BRD4 has been shown to associate with P-TEFb, affecting Pol 2 CTD phosphorylation, and hence, transcription elongation (*Itzen et al., 2014*).

These events coalesce ensuring a rapid remodeling of the inflammatory transcriptome, with hundreds of genes undergoing a dramatic upregulation (*Escoubet-Lozach et al., 2011*; *Chinenov et al., 2012*; *Gupte et al., 2013*; *Uhlenhaut et al., 2013*; *Tong et al., 2016*). Although essential for host defense, unabated inflammation imposes a threat to the host and can result in tissue damage and autoimmunity. One systemic mechanism that controls acute inflammatory response is a feedback loop whereby inflammatory cytokines trigger the production of steroid hormones known as glucocorticoids (GCs) (reviewed in [*Sacta et al., 2016*]). Lipophilic GCs diffuse through the cell membrane and bind the intracellular glucocorticoid receptor (GR), a transcription factor (TF), which then translocates to the nucleus and regulates gene expression. The transcriptional outcomes of GR activation are context-specific and are determined by the genomic GC response elements (GRE) to which the receptor binds. GR can bind directly to specific, usually pseudopalindromic, DNA sequences either as a homodimer or complexed with other TFs such as AP1 and STAT3 (*Biddie et al., 2011*; *Langlais et al., 2012*). In this context, GR recruits various coregulators such as members of the p160 family, HATs, the Mediator complex and ATP-dependent chromatin remodelers (*Weikum et al., 2017b*), ultimately leading to the activation of numerous genes including the anti-inflammatory genes, such as *Dusp1* and *Tsc22d3* (GILZ). At other sites, known as 'tethering' GREs, GR does not directly bind DNA but interacts with other DNA-bound TFs such as pro-inflammatory AP1 and NF-kB and usually represses their activity (reviewed in [*Chinenov et al., 2013*]) – a property fundamental to the ability of GCs to dramatically attenuate inflammation. In contrast to GR-mediated activation, the mechanisms of transcriptional repression by GR remain poorly understood. Strikingly, however, in a few cases analyzed, genes activated through Pol 2 recruitment and those induced by signal-dependent Pol 2 pause release were both susceptible to GR-mediated repression (*Gupte et al., 2013*).

Here, we use a combination of cell-based and genome-wide approaches to elucidate the mechanisms by which GR represses pro-inflammatory genes in primary macrophages challenged acutely with the TLR4 agonist lipopolysaccharide (LPS) and GCs. We present evidence of 'tethering' as a prevalent mechanism of repression among p65/GR co-regulated genes. We further demonstrate a widespread yet gene class-specific role of NELF in glucocorticoid-mediated repression of early elongation. Conversely, at other genes, GR precludes the ordered assembly of HATs, Brd4 and the Mediator complex which ultimately blocks Pol 2 recruitment and transcription initiation.

## Results

### Genomic binding of GR and p65 upon inflammatory and anti-inflammatory stimulation

To understand the mechanisms by which GR elicits its repressive effects, we first assessed by RNA-seq the global transcriptional changes upon acute activation of primary mouse bone-marrow-derived macrophages (BMDM) with LPS or LPS together with a synthetic GC dexamethasone (Dex) for 1 hr. At FDR < 0.1 we found that, compared to vehicle-treated BMDM, 597 genes were induced by LPS >1.5 fold. Of these, the induction of 201 genes was attenuated >1.3 fold by Dex co-treatment (*Figure 1A* and *Supplementary file 1*). As expected, GO analysis of acutely GR-repressed genes revealed predominantly those involved in cytokine signaling (*Figure 1A*).

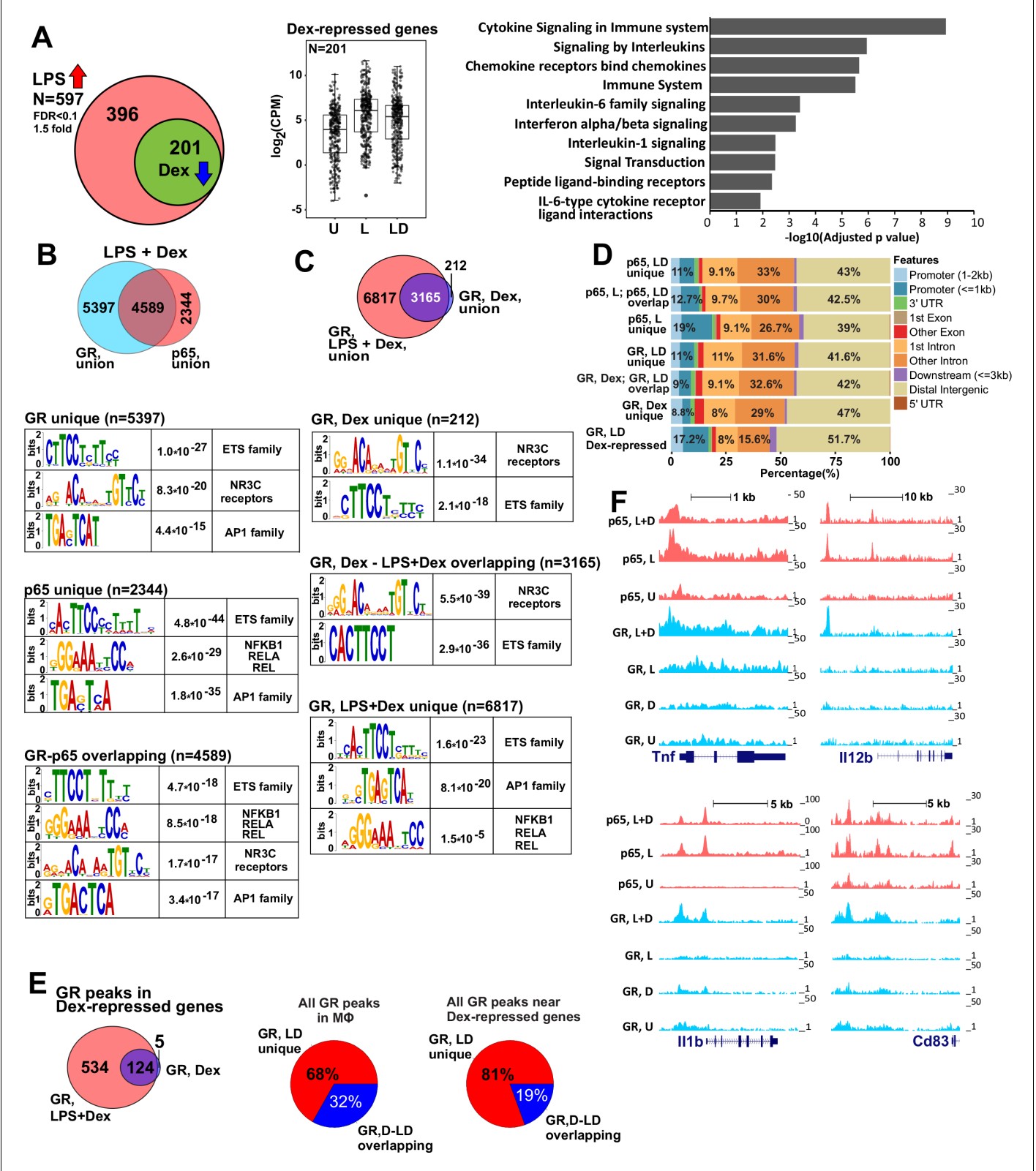

**Figure 1.** GR represses LPS-induced genes via p65-assisted tethering. (**A**) Over 30% of LPS-induced genes (597) in BMDM are repressed by Dex (201; Venn diagram and normalized expression values) and show a pro-inflammatory gene signature (GO analysis). BMDM were untreated (**U**) or treated with 10 ng/ml LPS ±100 nM Dex (**L and LD**) for 1 hr, and gene expression levels were determined by RNA-seq (n = 2). (**B**) The overlap between ChIP-seq peak calls for GR and p65 in LPS + Dex-treated BMDM (Venn diagram) was determined using *subsetByOverlap* function from GenomicRanges package

*Figure 1 continued on next page*

*Figure 1 continued*

(Bioconductor) with the minimum overlap of 1 nt (see Materials and methods). *Ab initio* sequence motif discovery and over-representation in each subset of GR or p65 binding peaks was determined using MEME-ChIP (*Ma et al., 2014*). E-values for the enrichment of the motif are shown. (**C**) Dex- and LPS + Dex-induced GR ChIP-seq peaks are shown (Venn diagram). LPS + Dex unique peaks are enriched for NF-kB-binding sites as indicated by MEME-ChIP analysis as in B. (**D**) Genomic location of p65 and GR binding sites relative to known genomic features is determined by *ChIPpeakAnno* (Bioconductor) (*Zhu et al., 2010*). (**E**) The distribution of GR-binding sites located in a 200 Kb region centered on LPS-induced Dex-repressed genes in BMDM treated with Dex or LPS + Dex (left). Pie-charts show the % of LD-unique GR peaks either genome-wide (center) or those associated with LPS-induced Dex-repressed genes only (right). (**F**) GR and p65 ChIP-seq read density profiles of representative LPS-induced Dex-repressed genes are shown for untreated (U), LPS (L) or LPS + Dex (L + D) treated BMDM. Also see *Figure 1—figure supplements 1–2* and *Supplementary files 1* and *2*.

DOI: https://doi.org/10.7554/eLife.34864.003

The following figure supplements are available for figure 1:

**Figure supplement 1.** Characterization of GR cistromes in Dex- and LPS + Dex-treated BMDM.

DOI: https://doi.org/10.7554/eLife.34864.004

**Figure supplement 2.** Characterization of p65 cistromes in LPS- and LPS +Dex treated BMDM.

DOI: https://doi.org/10.7554/eLife.34864.005

Despite rapid remodeling of the macrophage LPS-induced transcriptome in response to Dex observed by us and others (*Figure 1*, [*Rao et al., 2011*; *Chinenov et al., 2012*; *Uhlenhaut et al., 2013*; *Chinenov et al., 2014*]), no comprehensive analysis of the GR and p65 genome-wide occupancy under acutely repressing conditions has been reported. Therefore, we determined the distribution of p65 and GR genomic binding sites in BMDM pulsed with LPS, Dex or LPS + Dex for 45 min (see *Figure 1—figure supplements 1–2* and *Supplementary file 2* for quality metrics and comparison of replicates). Following LPS + Dex exposure, we detected 9987 GR peaks (union of two replicates) 5397 (54.1%) of which did not overlap with p65 peaks at the same conditions (*Figure 1B*, top, *Figure 1—figure supplement 1A*). Motif overrepresentation analysis in these GR unique peaks revealed predominance of centrally enriched NR3C-binding motifs, which represent GREs and highly related NR-binding sites, those for ETS family members, such as the macrophage lineage-determining TF SPI1 (PU.1 and SPIB), and AP1 family members (*Figure 1B*, *Figure 1—figure supplement 1B*, left panel). The analysis of p65 binding after LPS + Dex treatment revealed 7052 peaks (union of two replicates) of which 2344 (33.8%) were uniquely bound by p65 (*Figure 1B*, *Figure 1—figure supplement 2A*). Motif analysis indicated an enrichment of NF-kB/Rela binding motifs, as well as ETS and AP1 motifs (*Figure 1B*). Importantly, the GR and p65 cistromes shared 4589 peaks, which corresponds to nearly half of all GR- and 2/3 of all p65-binding peaks. Motif analysis of these peaks showed a predominance for NR3C/GRE, ETS family, NF-kB/Rela and AP1 binding motifs that were enriched near the peak summits (*Figure 1B*, bottom, *Figure 1—figure supplement 1B*, middle panel).

Because of the significant enrichment of peaks with NF-kB elements (especially among those overlapping p65-binding peaks) in the GR cistrome under repressing conditions, we performed GR ChIP-seq in BMDM treated with Dex only for 45 min to compare the two GR cistromes. In Dex-treated BMDM, GR-binding sites formed 3377 peaks. Of those, 3165 also appeared in the GR LPS + Dex cistrome (with only 212 peaks unique to Dex-treated BMDM), whereas 6817 were gained in the GR LPS + Dex cistrome (*Figure 1—figure supplement 1A*, right panel). ETS and NR3C-binding motifs were over-represented in both Dex-unique and Dex – LPS + Dex shared subsets of GR peaks and trended toward the peak summit (*Figure 1C*, *Figure 1—figure supplement 1B*, right panel). We did not detect NF-kB/Rela motif enrichment in these two subsets of GR-binding peaks. However, among 6817 peaks unique to the GR LPS + Dex cistrome we readily observed an overrepresentation of NF-kB and AP1 motifs while NR3C motifs were no longer enriched (*Figure 1C*, compare top/middle vs. bottom motif enrichment panels) indicating that inflammatory signaling and p65/NF-kB activation was driving GR recruitment to such sites specifically under repressing LPS + Dex conditions.

The majority of GR and p65-binding sites were located in distal intergenic (~39–47% of peaks) and intronic (~40% on average) regions (*Figure 1D*), similar to previously reported GR and p65 cistromes in various cell lines (*Reddy et al., 2009*; *Barish et al., 2010*).

To correlate GR binding with transcriptional outcomes, we focused on our subset of 201 LPS-induced Dex-repressed genes as determined by RNA-seq (*Figure 1A*, *Supplementary file 1*) and

evaluated GR peak localization within these genes and 100 Kb of their 5'- and 3'-flanking regions in Dex- and LPS + Dex-treated BMDM. In this subset, a somewhat larger fraction (~52%, compared to 39–47% genome-wide) of GR-binding peaks were located in distal intergenic regions, whereas the fraction of peaks in the introns dropped from 40% to 24% compared to whole-genome GR cistrome (*Figure 1D*). This shift was not due to a preponderance of shorter introns or genes in Dex-repressed subset (*Figure 1—figure supplement 1C*).

Comparison of GR binding near the 201 Dex-repressed genes with an entire GR cistrome shows that a greater fraction of binding sites was unique to the LPS + Dex condition (81% vs. 68%, *Figure 1E*) consistent with a disproportional increase in unique binding site utilization among this functionally constrained set of genes. Several representative examples of GR and p65 co-binding near GR-sensitive genes are shown in *Figure 1F*: at each gene, GR binding occurred at sites matching those of p65, but only in LPS + Dex and not LPS- or Dex-alone treated BMDM. Importantly, LPS-dependent p65 binding fully persisted in the presence of Dex. In fact, the total number of p65 binding peaks in the presence of LPS and LPS + Dex was comparable both genome-wide, and in the vicinity of our GR-repressed genes (*Figure 1—figure supplement 2A*, right and *2B*). In each case, ~2/3 of the LPS-induced p65 peaks persisted in LPS + Dex-treated BMDM. Moreover, among p65 LPS + Dex peaks functionally constrained to Dex-repressed genes, 80% (up from 68% genome-wide) overlapped LPS-induced peaks (*Figure 1—figure supplement 2C*). Interestingly, of the 201 genes repressed by Dex in the context of LPS-mediated macrophage activation, only 56 were repressed ≥1.3 fold (and only 16 of those ≥2 fold) upon treatment with Dex alone (*Supplementary file 1*; RNA-seq dataset from [*Chinenov et al., 2014*]) – further supporting a requirement for NF-kB activation for GR recruitment to the majority of genes Dex-sensitive genes. Combined, these results further corroborate a tethering model in which p65 is a central component of repression complexes in GC-treated BMDM.

## NELF mediates repressive effects of GR at paused genes

We have reported that at several pro-inflammatory genes in unstimulated BMDM, promoter-proximally paused Pol 2 is in a complex with NELF and enters productive elongation following LPS treatment (*Adelman et al., 2009*; *Gupte et al., 2013*). To assess how common this type of Pol 2 dynamics is among inflammatory genes, we performed Pol 2 ChIP-seq in untreated, LPS- or LPS + Dex treated BMDM.

*Figure 2A* shows Pol 2 tracks for six genes all of which were among 201 that were rapidly induced by LPS and repressed by Dex as established by RNA-seq (*Figure 1A*). Of those, *Tnf*, *Hilpda* and *Btg2*, all display accumulation of Pol 2 near the TSS in untreated BMDM. Upon a 45-min LPS treatment, we detect additional Pol 2 loading and, notably, its redistribution into the body of the gene; conversely, upon LPS + Dex treatment, Pol 2 largely remains near the TSS resembling a 'paused' pattern seen in the unstimulated BMDM (*Figure 2A*, left). In contrast, non-paused genes *Il1a*, *Il1b* and *Cd83* display no substantial Pol 2 occupancy in the unstimulated BMDM, and a dramatic and uniform increase in Pol 2 occupancy throughout the gene in response to LPS, which is nearly abrogated by co-treatment with Dex (*Figure 2A*, right).

These findings raised the possibility that GR mediates its repressive effects genome-wide by regulating distinct steps of Pol 2 transcription cycle depending on the rate-limiting step for gene activation. To address this possibility, we first calculated Pol 2 pausing indexes (PI) for approximately 300 transcripts corresponding to our 198 LPS-induced Dex-repressed genes (three genes were excluded due to the conflict of annotation). As described in *Nechaev et al. (2010)*, we defined PI as the ratio of log-transformed normalized Pol 2 counts around the promoter (−200/+500 bp relative to the annotated TSS) to those within the gene body downstream of +500 bp (*Figure 2B*, *Supplementary file 3*). Based on the PI in untreated BMDM, we classified GC-repressed genes into two groups: 61 transcripts had a PI >1 and were considered to be paused (twice as much of Pol2 at the promoter region versus gene body), whereas 82 had a PI <0.8 and were considered non-paused (see Materials and methods and [*Nechaev et al., 2010*]). *Figure 2C* shows Pol 2 distribution within the −200/+1500 region for individual transcripts of both classes in unstimulated BMDM, as well as BMDM exposed for 45 min to LPS or LPS + Dex. The read density distribution for 61 paused and 82 non-paused transcripts in differentially treated BMDM (*Figure 2D*) reveals a peak of Pol 2 occupancy in the promoters of the paused genes, additional Pol 2 loading, and, importantly, its entry into gene bodies in response to LPS. Co-treatment with Dex decreases Pol 2 occupancy in gene body with

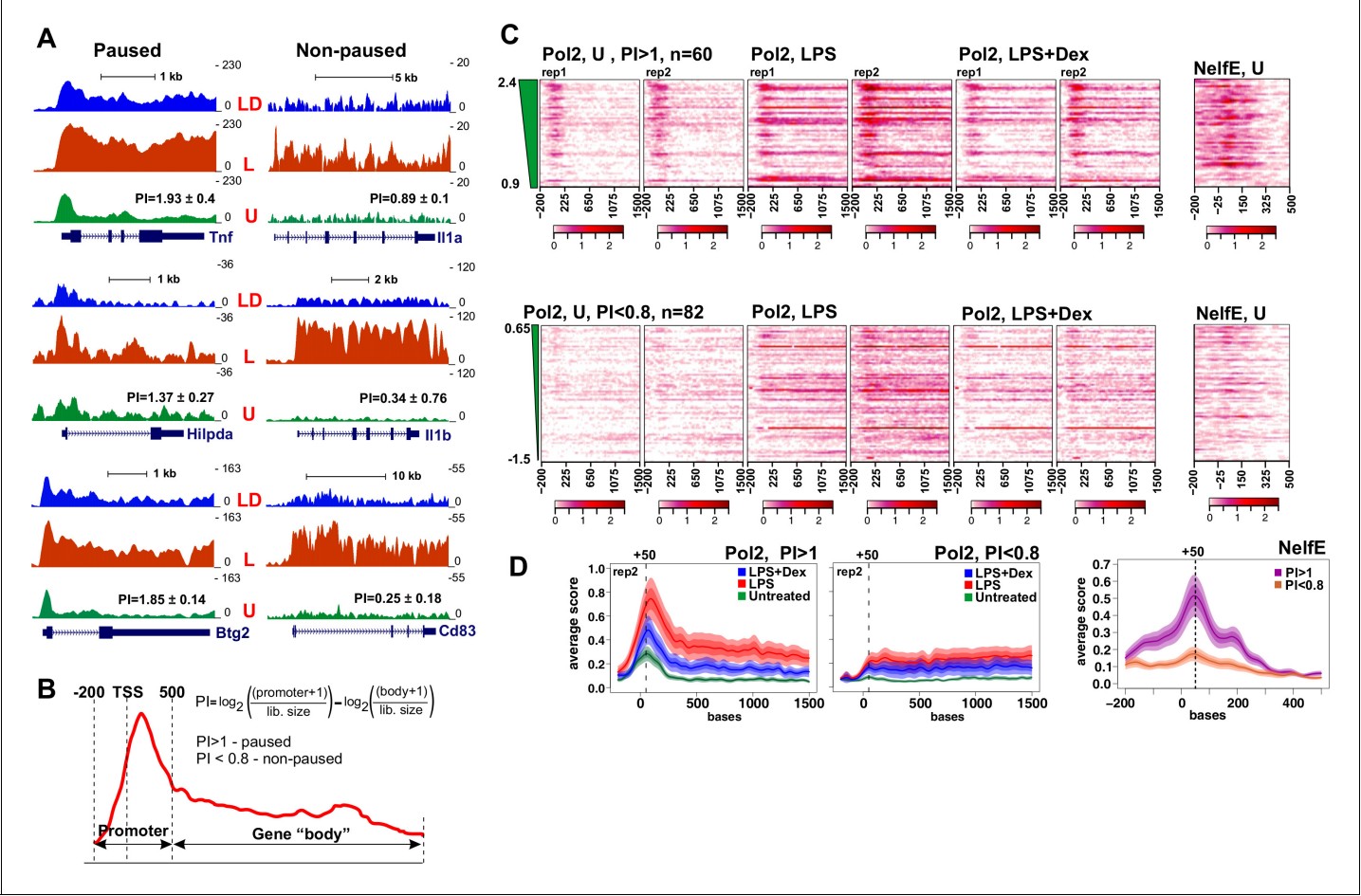

**Figure 2.** Pol 2 and NELF dynamics at different classes of GR-sensitive genes. (**A**) Pol 2 ChIP-seq read density profiles and pausing indexes (PI) for representative paused and non-paused genes in the untreated (U), LPS (L) and LPS +Dex (LD) treated BMDM. (**B**) PI (a ratio of Log-transformed Pol 2 counts at the promoter and gene body in untreated BMDM) was calculated for all LPS-induced Dex-repressed transcript variants with unique 5′ ends (see Materials and methods). Genes with PI >1 were considered paused and those with a PI <0.8 non-paused. (**C**) Pol 2 ChIP-seq heat maps of paused (n = 62) and non-paused (n = 82) transcripts sorted by the PI indexes corresponding to 198 Dex-repressed genes (see Materials and methods) are shown for the U, L and L + D conditions for each individual replica. Only transcripts overlapping Pol 2 peaks in LPS-treated BMDMs as determined by MACS2 are shown. NELF-E heat maps from U BMDM ChIP-seq for the same transcript classes are shown on the right. Heat maps scales are equalized to visualize Pol 2 and NELF distribution across the genes; color scale bars are shown below corresponding maps. (**D**) Average Pol 2 (in each treatment condition) and NELF-E (untreated BMDM) occupancy for each gene class defined in C. The confidence band shows the SEM and 95% confidence interval. Also see *Supplementary files 2* and *3*.

DOI: https://doi.org/10.7554/eLife.34864.006

most Pol 2 remaining near the TSS (*Figure 2C and D*). Conversely, little Pol 2 is seen in the non-paused genes in untreated BMDM; Pol 2 occupancy increases dramatically throughout the genes in LPS-treated BMDM and this loading is largely abrogated by Dex (*Figure 2C and D*), consistent with the pattern shown in *Figure 2A* for representative genes.

Because Pol 2 pausing within the first 100 nt of a gene is mediated by NELF (*Adelman and Lis, 2012*), we assessed genome-wide NELF distribution by ChIP-seq using antibodies to the NELF-E subunit of the complex. Aligned with Pol 2 PI heat maps, NELF-E occupancy matched closely Pol 2 distribution in untreated BMDM with striking accumulation immediately downstream of TSS of paused genes and little to no NELF-E seen in non-paused genes (*Figure 2C and D*, far right). Indeed, read density distribution in NELF-E ChIP-seq shows highly gene class-specific NELF-E occupancy at paused (PI >1) promoters (*Figure 2D*, right).

As reported previously for a few individual genes (*Adelman et al., 2009*; *Schaukowitch et al., 2014*), following LPS stimulation, NELF-E was broadly evicted from promoters of LPS-induced genes

with little occupancy detected at 1 hr (*Figure 3A*). Interestingly, however, this dismissal was transient, as despite continued LPS exposure, NELF reloaded onto promoters reaching widespread occupancy by 3 hr (*Figure 3A*, also see average occupancy graphed for all paused transcripts). This release and reloading can be seen at specific paused GC-sensitive genes, *Tnf*, *Hilpda*, and *Btg2* (*Figure 3A*, right), which show substantial NELF-E occupancy at the TSS co-localizing with Pol 2 peaks in resting BMDM, its dissociation following a 1 hr LPS induction, and re-establishment of the TSS-associated NELF-E peaks by 3 hr.

To directly assess whether NELF occupancy in GC-sensitive genes in BMDM correlates with Pol 2 pausing in early elongation, we compared the NELF-E and Pol 2 cistromes in the unstimulated BMDM. Among the LPS-induced Dex-sensitive genes with PI >1 (approximately 24% of 300 Dex-repressed transcripts), 86.3% displayed promoter-associated NELF-E peaks, compared to only 31.7% in genes with PI <0.8 (which comprised approximately 66% of 300 transcripts; *Figure 3B*, left). Importantly, similar relative numbers of paused and non-paused genes (23% and 70%, respectively; *Figure 3—figure supplement 1B*) were found among LPS-induced Dex-insensitive genes from RNA-seq (*Figure 1A*). In this group, NELF-E occupancy in untreated BMDM was again much more prevalent in paused genes (81.1%) than in non-paused ones (44.2%). Thus, GR does not preferentially repress genes in one class vs. the other, and high levels of TSS-associated NELF in a basal state is a common feature of paused genes irrespective of their sensitivity to GC.

Given that NELF and Pol 2 co-localize at the TSS of the paused genes in unstimulated BMDM, that activation of such genes by LPS coincides with NELF dismissal, and that Pol 2 remains near the promoters of these genes under repressing conditions consistent with their early elongation arrest, we questioned whether GR-mediated repression was globally mediated by NELF. We first evaluated NELF-E occupancy in BMDM co-treated with LPS + Dex by ChIP-seq and found the relative distribution of peaks among paused (PI >1) and non-paused (PI <0.8) repressed genes to be indistinguishable from NELF-E distribution in resting BMDM (83.6% and 33.2%, respectively; *Figure 3B*, right – compare to left). We then evaluated NELF-E distribution across several of our target genes in the presence of LPS + Dex and detected striking promoter-proximal NELF peaks in paused *Tnf*, *Myc*, *Errfi1* and *Ccl2*, but not in non-paused *Il1b* or *Lif* (*Figure 3B*). To address directly whether NELF is necessary for GR-mediated repression, we used a new mouse strain conditionally lacking the NELF-B subunit and, hence, the functional NELF complex in myeloid cells (see Materials and methods). BMDM from NELF-B LysM-Cre mice (NELF-B KO) show a dramatic reduction in NELF-B mRNA and protein (*Figure 3C*, top). Importantly, as the NELF complex requires all four subunits for stability and the loss of a single subunit leads to the proteolytic degradation of the complex (*Gilchrist et al., 2008*), immunoblot also reveals a near complete loss of the NELF-E protein in the BMDM of the NELF-B KO (*Figure 3C*, top). Using WT and NELF-B KO BMDM, we then compared GR-mediated repression of our candidate GC-sensitive genes. Consistent with the lack of overt phenotype in these mice, RNA-seq of resting BMDM of the two genotypes revealed no significant differences in the expression levels of LPS-induced Dex-repressed genes at baseline (*Figure 3—figure supplement 1C*). Moreover, at the time-frame examined, LPS challenge led to a similar induction of *Tnf*, *Myc*, *Errfi1*, *Ccl2*, *Il1b* and *Lif* transcripts irrespective of the genotype (*Figure 3C*, bottom left). Interestingly, for all genes classified as 'paused', repression by Dex was significantly attenuated in the NELF-B KO BMDM, but not in non-paused genes *Il1b* and *Lif* (*Figure 3C*, bottom right). Collectively, these findings strongly suggest that NELF-mediated block in productive elongation is an integral part of GR-mediated repression of paused genes.

To extend these observations to a whole-genome level, we analyzed transcriptomes from the WT and NELF-B KO BMDM treated with LPS + Dex for 1 hr by RNA-seq which identified 393 differentially expressed genes (fold change = 1.5, FDR p<0.05). Out of 201 genes that were repressed by Dex in the WT BMDM (*Figure 1A*), 23 were expressed at higher level in the LPS + Dex-treated NELF-B KO BMDM; notably, 21 of them had PI >0.8 (*Figure 3D*). Conversely, out of 396 LPS-induced Dex-insensitive genes, only nine were upregulated in the LPS + Dex-treated NELF-B KO BMDM, 7 of which had PI >0.8 (*Figure 3—figure supplement 1D*, left). These observations indicate that NELF ablation disproportionally affects paused LPS-induced Dex-repressed genes.

Because NELF release is triggered by CDK9-mediated phosphorylation, we evaluated the recruitment of CDK9 to the TSS of paused and non-paused genes. Consistent with earlier observations (*Luecke and Yamamoto, 2005*), GR inhibited LPS-induced CDK9 recruitment but did so irrespective

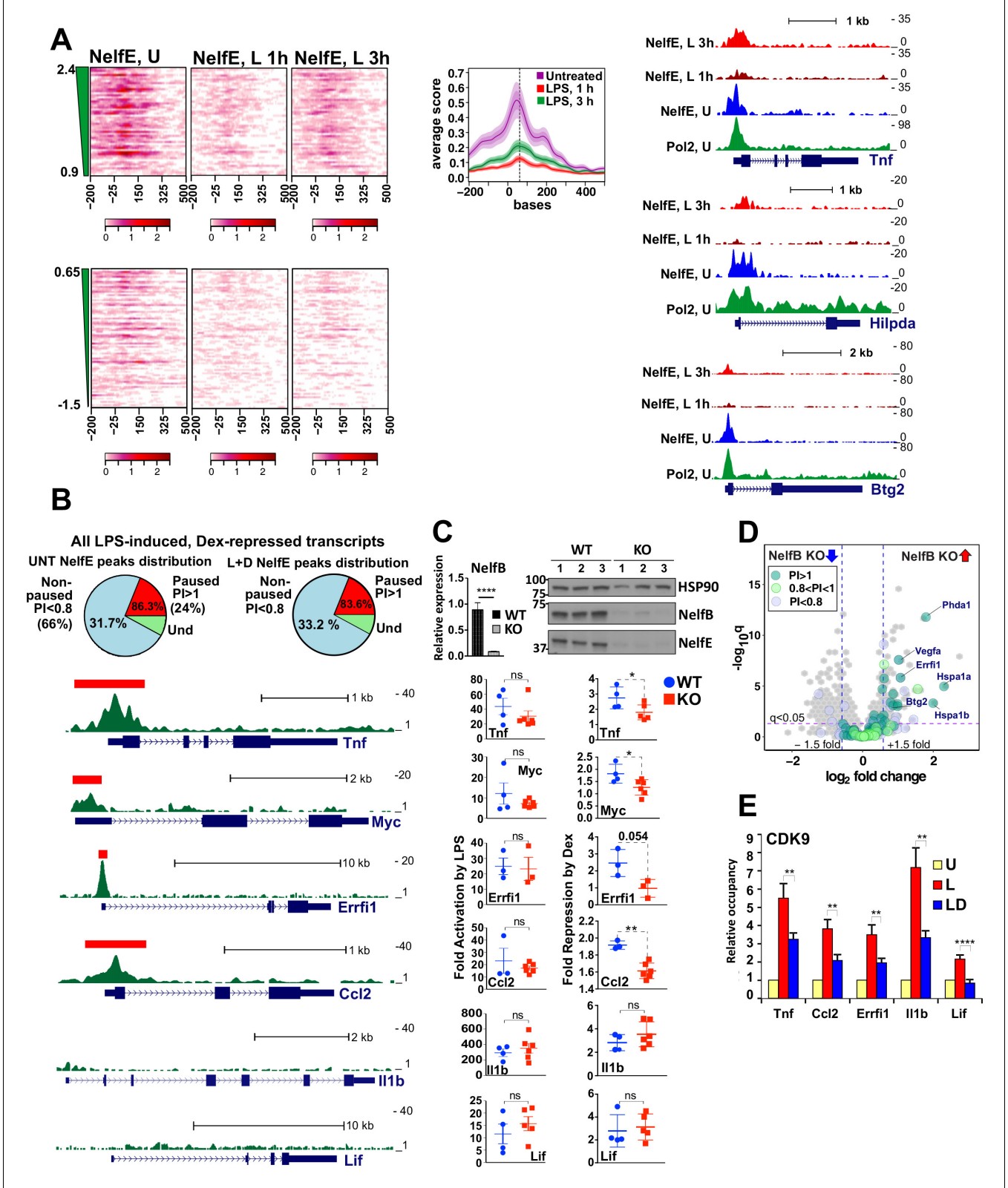

**Figure 3.** Gene-class-specific contribution of NELF to GR-mediated repression. (A) Heat maps show NELF-E occupancy in the U (from *Figure 2C*), 1 and 3 hr L-treated BMDM for paused and non-paused transcripts. Average occupancy for the paused genes in each condition is graphed as in *Figure 2D*. Representative examples of Pol 2 and NELF ChIP-seq read density profiles are shown on the right. (B) Pie charts show the percentage of all paused (24%) and non-paused (66%) LPS-induced Dex-repressed genes that exhibit promoter-proximal NELF-E binding in the UNT (86.3% and 31.7%, *Figure 3 continued on next page*

*Figure 3 continued*

respectively) and L + D conditions (83.6% and 33.2%, respectively). NELF-E ChIP-seq read density profiles for the L + D condition are shown for a set of representative genes. Red rectangles in *Tnf*, *Myc*, *Errfi1* and *Ccl2* profiles indicate MACS2 NELF-E peaks in the L + D condition. (C) NELF-B KO mice were generated as described in Methods. NELF-B RNA in WT and KO BMDM was quantified by RT-qPCR and normalized to *Actb* (n = 5, p<0.0001, two-tailed Student's t-test; error bars are SEM). For western blots, three mice per genotype were used to visualize NELF-B, NELF-E and HSP90 as a loading control (top). Bottom: WT and NELF-B KO BMDM were U or treated with L-/+D for 30 min (*Tnf*) or 1 hr (all others) and the expression of indicated genes (matching those in B) was assessed by RT-qPCR, normalized to *Actb*, and shown as 'fold activation by LPS' over basal levels (=1) and 'fold repression by Dex' (a ratio of L over L + D level of each transcript). *p<0.05, **p<0.01 (Two-tailed Student's t-test). Error bars are SEM. (D) The volcano plot comparing gene expression in L + D (1 hr) treated BMDM from the WT vs. NELF-B KO mice (n = 3) (fold change = 1.5, FDR p<0.05). Pausing indices (PI) of 201 LPS-induced Dex-repressed genes from *Figure 1A* are shown in color. (E) CDK9 occupancy at selected genes in BMDM treated for 1 hr as indicated. n = 4–9. **p<0.01, ****p<0.001 (two-tailed Student's t-test). Error bars are SEM. Also see *Figure 3—figure supplement 1* and *Supplementary file 2*.

DOI: https://doi.org/10.7554/eLife.34864.007

The following source data and figure supplements are available for figure 3:

**Source data 1.** Source raw data for Fig. 3C (RT-qPCR in WT and NELF-B KO) and 3E (ChIP-qPCR for CDK9).

DOI: https://doi.org/10.7554/eLife.34864.009

**Figure supplement 1.** Characterization of Pol 2 and NELF cistromes in BMDM.

DOI: https://doi.org/10.7554/eLife.34864.008

**Figure supplement 1—source data 1.** RNA-seq baselines in wt vs NELF-B KO.

DOI: https://doi.org/10.7554/eLife.34864.010

of the gene class (*Figure 3E*) suggesting that NELF retention rather than CDK9 occupancy serves as a defining class-specific feature of glucocorticoid repression of paused genes.

## GR-mediated repression of non-paused genes results in attenuation of histone H4 acetylation and BRD4 binding

The dynamics of Pol 2 binding at non-paused genes, as shown in *Figure 2*, suggested that the major barrier to activation at these genes is the loading of Pol 2. BMDM surpass this barrier by recruiting NF-kB and AP1 to enhancer regions (*Glass and Natoli, 2015*) that in turn assemble histone-modifying proteins, which help create a more permissive chromatin environment for the binding of basal transcriptional machinery and Pol 2. In particular, histone tail modifications, which are associated with both enhancers and promoters are H3K9Ac and H4K5/8/12Ac (*Smale et al., 2014*). Because these marks correlate with gene transcriptional status, we evaluated the histone acetylation at a subset of our GC-repressed genes of each class.

We noted striking differences in histone tail modifications between representatives of the two gene classes. In particular, paused genes - *Tnf* and *Ccl2* - contained high basal levels of H4PanAc and, specifically, H4K5Ac, at both TSS- and kB-binding sites which were unaffected by LPS or LPS + Dex treatment (*Figure 4A*, bottom row). In contrast, non-paused genes - *Il1b* and *Il1a* - showed a significant increase in H4Ac levels only after LPS treatment, especially at the *Il1b* TSS and two *Il1a* kB enhancers at −10 Kb and −20 Kb, and this increase was fully attenuated by Dex (*Figure 4A*, top row).

The change in acetylation seen preferentially at our non-paused genes, appeared to denote a specific 'histone code' for histone binding proteins that could potentially affect the transcription of these genes. In particular, BRD4, the Bromodomain and Extra-Terminal domain (BET) histone binding protein, affects inflammatory cytokine transcription both in vitro and in vivo through direct binding to acetylated H3 and H4 (*Shi and Vakoc, 2014*). The changes in H4PanAc including H4K5/K12Ac in several GC-sensitive genes, suggested a possible role for BRD4 in transcriptional repression by GR. To test this hypothesis, we first assessed activation of pro-inflammatory genes by LPS in the presence of increasing concentrations of I-BET, an inhibitor of BRD4 binding. The induction of *Il1b* and *Il1a* transcripts was significantly attenuated by I-BET in a dose-dependent manner, whereas *Tnf* and *Ccl2* induction persisted (*Figure 4B*). In agreement with gene expression results, ChIP-qPCR experiments revealed that BRD4 was recruited to promoters of non-paused genes *Il1b* and *Il1a* upon LPS treatment and, interestingly, this recruitment was attenuated by Dex (*Figure 4C*). Conversely, at the paused genes, *Tnf* and *Ccl2*, BRD4 was readily detectable at the TSS in unstimulated BMDM and this association did not significantly change after either LPS or LPS + Dex treatment. Thus, BRD4

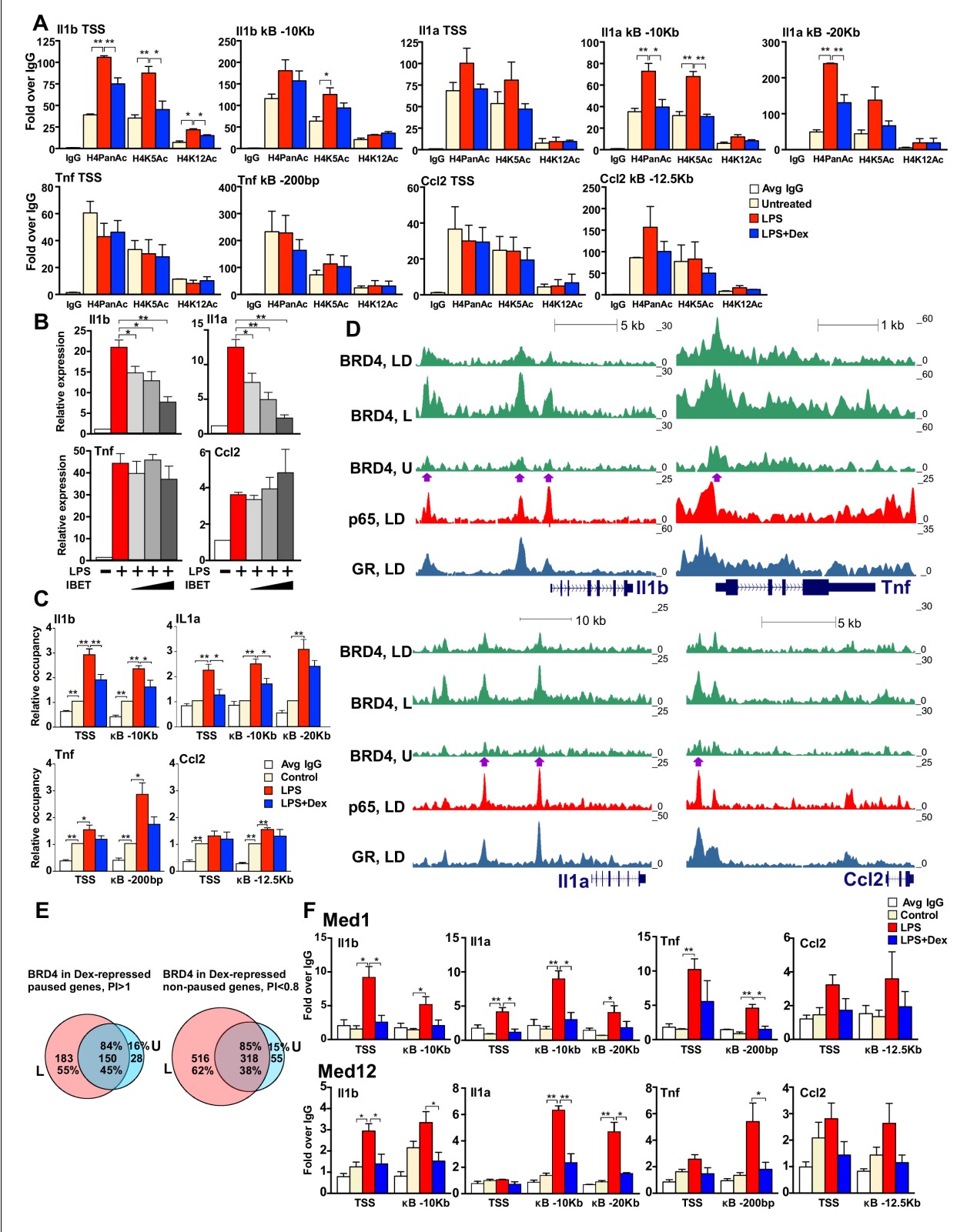

**Figure 4.** GR inhibits H4 acetylation, BRD4 and Mediator assembly at non-paused genes. (**A**) BMDM were treated as indicated, and H4PanAc, H4K5Ac and H4K12Ac at the TSS and indicated kB sites were assessed by ChIP. qPCR signals were normalized to r28S gene and expressed as relative enrichment over normal IgG (=1). A two-tailed Student's t-test was used for comparing means (n ≥ 3; *p<0.05, **p<0.01). Error bars are SEM. (**B**) BMDM were pre-treated with I-BET (10 nM, 100 nM, 1 μM) for 30 min followed by addition of LPS for 30 more min. Gene expression was assessed by

*Figure 4 continued on next page*

Figure 4 continued

RT-qPCR and normalized to that of Actb. A two-tailed Student's t-test was used for comparing means (n ≥ 3; *p<0.05, **p<0.01). Error bars are SEM. (C) BRD4 occupancy was assessed by ChIP-qPCR as in A with IgG ChIP as a background metric and expressed as relative enrichment over untreated for each site (=1). A two-tailed Student's t-test was used for comparing means (n ≥ 3, *p<0.05, **p<0.01). (D) ChIP-seq read density profiles for BRD4, GR and p65 in the U, L or L + D treated BMDM. Purple arrows indicate peaks specifically noted in Results. (E) Venn diagrams show overlapping BRD4 peaks for Dex-repressed paused and non-paused genes in the U and L condition. Overlapping peaks were determined as described in *Figure 1* and Materials and methods. (F) Med1 and Med12 occupancy is analyzed by ChIP-qPCR as in A (n ≥ 3). Also see *Figure 4—figure supplement 1* and *Supplementary file 2*.

DOI: https://doi.org/10.7554/eLife.34864.011

The following source data and figure supplement are available for figure 4:

**Source data 1.** Source raw data for *Figure 4A, C, F* (ChIP-qPCR for H4Ac, Brd4 and Mediator) and 4B (RT-qPCR).
DOI: https://doi.org/10.7554/eLife.34864.013

**Figure supplement 1.** Characterization of BRD4 cistromes in BMDM.
DOI: https://doi.org/10.7554/eLife.34864.012

occupancy patterns at the promoters of these genes resembled signal-responsive H4Ac profiles suggesting that loss of BRD4 in response to Dex may underlie GR-mediated repression of, specifically, the non-paused genes.

We then assessed genome-wide distribution of BRD4 by ChIP-seq. Not surprisingly, we observed frequent BRD4 binding across the genome in untreated BMDM (~3700 peaks, *Figure 4—figure supplement 1A*, left panel). There was a 1.8-fold increase in the number of BRD4 peaks in response to LPS relative to that in untreated BMDM (4345 new peaks, *Figure 4—figure supplement 1A*, left panel). The increase in the total peak number was even more apparent when limited to LPS-induced genes: 2.9-fold for Dex-insensitive or -repressed genes (*Figure 4—figure supplement 1A*, middle and right panel, respectively). Furthermore, BRD4 loading density specifically at our Dex-sensitive genes increased dramatically in response to LPS which, interestingly, was nearly abrogated by Dex - a trend very apparent at promoters, but also significant at BRD4:p65 shared binding sites (*Figure 4—figure supplement 1B*). BRD4 read distribution at individual non-paused genes of interest reflected this dynamics. For example, the *Il1b* TSS and −2.3 Kb and −10 Kb p65 enhancers acquired strong BRD4 binding in response to LPS which was significantly attenuated by Dex, concomitantly with GR loading (*Figure 4D*, left top, purple arrows). *Il1a* also displayed increased LPS-induced BRD4 loading at kB-associated upstream enhancers (−10 Kb and −20 Kb) with a dramatic reduction in occupancy upon Dex co-treatment corresponding to GR binding at both regions (*Figure 4D*, left bottom, purple arrows). Consistent with our ChIP-PCR data, paused genes, *Ccl2* (−12.5 kB enhancer) and especially *Tnf* (TSS) exhibited modest yet detectable BRD4 occupancy in untreated BMDM that was potentiated by LPS but only minimally affected by Dex (*Figure 4D*, right). Moreover, our analysis of BRD4 occupancy at Dex-sensitive genes of the two classes revealed that in paused genes, 45% of the BRD4-binding sites seen in LPS-treated BMDM were already pre-bound in untreated cells and 55% were LPS-induced; in non-paused genes, however, only 38% of the sites were pre-occupied in untreated BMDM, whereas 62% were LPS-dependent (*Figure 4E*). Thus, our functional studies together with occupancy data suggest that the activation of non-paused genes is more dependent on BRD4 recruitment, and therefore, its dismissal may have a greater impact on genes of this class.

Initial BRD4 characterization revealed its interaction with the Mediator complex subunits MED1 and MED12 (*Jang et al., 2005*; *Lovén et al., 2013*). Mediator is an evolutionarily conserved multiprotein co-activator complex that facilitates transcriptional activation of many genes in part by linking physically and functionally effector TFs and Pol 2. In the context of LPS-induced activation of pro-inflammatory genes, MED1 is reportedly recruited to both the TSS and p65 enhancers (*Hargreaves et al., 2009*; *Brown et al., 2014*), occupying similar sites across the genome as BRD4, and the two appear to stabilize each other's occupancy at enhancer regions (*Lovén et al., 2013*). We therefore assessed MED1 and MED12 occupancy at the promoters and p65 enhancers of GR-sensitive genes and found that both were recruited to TSS and p65 sites in response to LPS treatment and their recruitment was attenuated by Dex (*Figure 4F*). Thus, by inhibiting BRD4 binding to the TSS and certain enhancer regions at non-paused genes, GR destabilizes MED1 and MED12 occupancy ultimately affecting Pol 2 recruitment. Of note, MED1 and MED12 loss in response to Dex

occurred at paused genes as well (*Figure 4F*), suggesting that GR may antagonize the Mediator complex binding irrespective of its effects on BRD4.

## GR attenuates histone acetylation, cofactor assembly and Pol 2 recruitment to non-paused genes by blocking the recruitment of p300

GR activation disrupted histone acetylation and subsequent BRD4 and Mediator complex assembly at our analyzed non-paused genes, suggesting a central role for LPS-induced histone acetylation as a potential target for GR. Various HATs, including GCN5 and p300, have been implicated in altering modifications at the histone H3 and H4 tails (*Smale et al., 2014*) Furthermore, p300 has been shown to also interact with and acetylate p65, which contributes to the activation of NF-kB-dependent genes (*Huang et al., 2009*; *Nagarajan et al., 2014*; *Roe et al., 2015*). Thus, p300 appeared as a plausible HAT to execute H3/H4 acetylation, thereby dictating the recruitment of BRD4 to the promoters and kB sites of our genes of interest. ChIP-qPCR experiments revealed LPS-dependent recruitment of p300 to the TSS- and p65-binding sites of non-paused genes *Il1a* and *Il1b*, which was blocked by Dex. Interestingly, at paused genes, p300 occupancy varied, showing some LPS-potentiated yet Dex-insensitive recruitment to *Tnf*, but a strong constitutive occupancy at *Ccl2* (*Figure 5A*). Notably, loss of p300 from these genes did not reflect a simple 'titration' of p300 by the activated GR potentially broadly sequestering it away from kB enhancers, as p300 occupancy at the p65-binding sites of LPS-induced Dex-insensitive genes identified by our RNA-seq analysis - *Cxcl10*, *Cd40*, *Tnfsf9*, *Trim13* - was fully resistant to Dex (*Figure 5B*).

We reasoned that p300 catalytic activity rather than its occupancy is a better indicator of whether or not this HAT is involved in regulating target GR-sensitive genes. Therefore, a selective and competitive inhibitor of the p300 HAT activity, C646, was used to determine whether p300-mediated acetylation of histones was necessary for transcriptional activation of candidate pro-inflammatory genes. C646 attenuated in a dose-dependent manner LPS-mediated induction of non-paused genes *Il1b* and *Il1a*, whereas activation of paused genes *Tnf* and *Ccl2* was unaffected (*Figure 5C*), consistent with a selective requirement for p300 at the non-paused genes. Furthermore, if GR represses *Il1a* and *Il1b* specifically by precluding p300 recruitment, its ectopic introduction into cells should rescue LPS-mediated induction irrespective of GC treatment. *Figure 5D* shows that overexpression of wild-type p300 but not its ΔHAT mutant devoid of the catalytic activity in macrophage-like RAW264.7 cells dramatically and specifically reversed GR-mediated repression of non-paused genes. This suggests that GR represses these genes by precluding p300 recruitment, H3/H4 acetylation and the assembly of the BRD4-Mediator complex, ultimately blocking Pol 2 loading.

## Discussion

Despite the unmatched therapeutic utility of GCs stemming in large part from rapid and direct transcriptional repression of the key inflammatory genes, our knowledge of the overall architecture, dynamics, stability and distribution of such repressive GR complexes in inflammatory cells has been lacking. Given fundamental differences in the rate-limiting events for inflammatory gene activation, we sought to dissect the mechanisms by which GR elicits repression in such distinct gene classes and use genome-wide approaches to assess the generality of our findings.

Numerous studies in cell culture and cell-free systems implicated physical interactions between GR, NF-kB and AP1 family members in the inhibition of pro-inflammatory gene transcription (reviewed in [*Sacta et al., 2016*]) and indeed, we observe extensive co-localization of GR and the NF-kB subunit, p65, genome-wide and especially nearby Dex-repressed genes following short-term LPS + Dex co-treatment – conditions under which we observe rapid glucocorticoid repression. GCs did not cause global displacement of p65; in fact, the number on p65-binding sites in the presence of LPS vs. LPS + Dex is comparable. Moreover, 80% of p65 peaks associated with our Dex-repressed genes overlap in LPS- and LPS + Dex-treated BMDM. Interaction with p65 is further corroborated by the persistence of p65 peaks near our candidate Dex-repressed genes of both classes. With respect to GR binding, both globally and restricted to Dex-repressed genes, several observations point to a tethering mechanism. First, the predominant motifs enriched in GR peaks present uniquely under LPS + Dex conditions are those of NF-kB and AP1 and not the NR3C motif overrepresented in Dex-treated BMDM or peaks shared between the two cistromes. Second, when compared between an entire genome and restricted to Dex-repressed genes, the fraction of LPS + Dex unique GR-binding

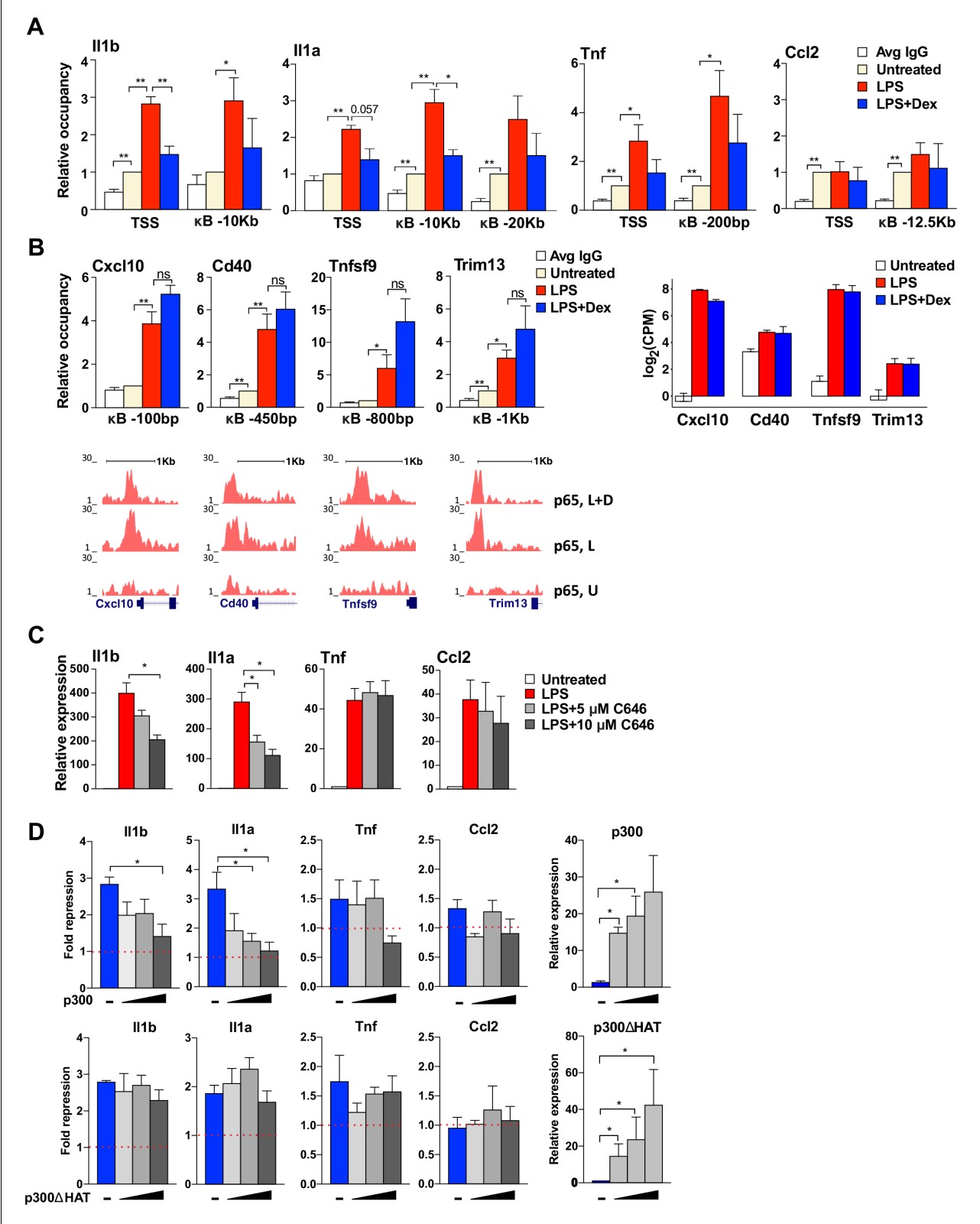

**Figure 5.** GR-mediated repression of non-paused genes is associated with the diminished p300 function. (**A**) p300 occupancy at indicated kB-binding sites is evaluated as in *Figure 4C* (n ≥ 3). (**B**) p300 occupancy at indicated kB binding sites is evaluated as in A (n ≥ 3; top panel). p65 ChIP-seq read density distribution in U-, L- or L + D-treated BMDM for corresponding kB-binding sites is shown (bottom panel). Expression level (log(CPM) values) for LPS-induced Dex-insensitive genes as determined by RNA-seq in *Figure 1A* for the WT BMDM (untreated, LPS 1 hr, L + D 1 hr, n = 2, right panel). (**C**)

*Figure 5 continued on next page*

*Figure 5 continued*

BMDM were treated with LPS for 30 min followed by addition of 5 μM or 10 μM C646 for another 1 hr. The expression of indicated genes was assessed as described in *Figure 4B* (n ≥ 3). (**D**) RAW264.7 cells were transfected with increasing amounts of pcDNA3-p300 or pcDNA3-p300(ΔHAT) (0, 50, 100 and 150 ng/well) as described in Materials and methods. Cells were treated with 100 ng/ml LPS ±100 nM Dex for 1 hr. Gene expression was analyzed as described in *Figure 3C* (n ≥ 3).

DOI: https://doi.org/10.7554/eLife.34864.014

The following source data is available for figure 5:

**Source data 1.** Source raw data for *Figure 5A-B* (p300 ChIP-qPCR) and 5C-D (RT-qPCR).
DOI: https://doi.org/10.7554/eLife.34864.015

sites is increasing substantially from 68% to 81%. Third, the majority of the 201 Dex-sensitive genes are only repressed in LPS-activated macrophages, pointing to a requirement for NF-kB activation for GR recruitment. Indeed, the analysis of GR occupancy nearby our candidate Dex-sensitive genes of both classes reveals co-localized GR and p65 peaks associated with NF-kB enhancers under repressing LPS + Dex conditions and no GR binding in Dex-only - treated macrophages. Thus, although this is certainly not the only mechanism by which GR affects inflammatory gene expression (*Rao et al., 2011*; *Uhlenhaut et al., 2013*; *Oh et al., 2017*; *Weikum et al., 2017a*), tethering to p65 is a widespread regulatory mechanism that GR relies upon to elicit acute repression of pro-inflammatory genes in macrophages.

How GR enacts repression appears to depend on the state of the target promoters prior to activation. At paused genes, Pol 2 is pre-loaded, bound by NELF and 'stalled' nearby the TSS, (*Gilchrist et al., 2012*). These genes have elevated levels of histone acetylation at the TSS, suggestive of an open chromatin state, which would favor constitutive Pol 2 loading and transcription initiation. Conversely, non-paused genes show little Pol 2 occupancy in unstimulated BMDM. Among our Dex-repressed genes, both classes were well represented: in a set of transcripts filtered for Pol 2 occupancy and used to calculate PI, 61 were paused and 82 were not; in a total pool of transcripts corresponding to LPS-induced Dex-repressed 198 genes, approximately 24% were paused (PI > 1) and 66% non-paused (PI < 0.8). This distribution matched closely that of genes that were LPS-induced but insensitive to Dex (23% and 70%, respectively), suggesting that GR does not display a preference for a specific gene type for repression.

Given a critical role of NELF in establishing Pol 2 pausing (*Gilchrist et al., 2008*; *Core et al., 2012*), we evaluated the genomic distribution of NELF at our LPS-induced Dex-repressed genes in basal, activated and repressed state. This analysis revealed a striking correlation between Pol 2 promoter-proximal pausing and NELF occupancy. Indeed 81% of the paused genes had TSS-associated NELF peaks compared to only 44% on non-paused genes. As expected, NELF dissociated from Pol 2 after LPS treatment, presumably due to P-TEFb-mediated phosphorylation, enabling productive elongation. Although the rate of NELF dismissal varies depending on culture conditions and in our experience takes 30–60 min, this loss is consistently transient as NELF 're-loads' onto the TSS of these genes despite continuous presence of LPS. We previously reported a highly dynamic occupancy of NELF at the *Tnf* promoter (*Adelman et al., 2009*), but a global synchronous reloading of NELF onto promoters of activated pro-inflammatory genes was unexpected. We envision that NELF re-loading may provide a tonic control of the inflammatory response by limiting further entry of Pol 2 into productive elongation (*Aida et al., 2006*), yet maintain genes poised for induction by preserving a nucleosome-depleted environment (*Gilchrist et al., 2008*; *Core et al., 2012*). A distinct mechanism of 'tonic control' of inflammatory gene expression was recently described for a transcriptional repressor Hes1 which limits the recruitment of P-TEFb and hence, NELF release and Pol 2 elongation (*Shang et al., 2016*). In that regard, it would be informative to examine the dynamics of P-TEFb and phosphorylation of the Pol 2 CTD at the promoters of these genes over the time frame of NELF recycling. Interestingly, paused genes were originally proposed to be fast and transient responders to inducing signals (*Adelman et al., 2009*; *Rogatsky and Adelman, 2014*); NELF reloading despite prolonged LPS exposure could potentially contribute to cessation of activation and establishing a 'tolerant' LPS-unresponsive state. More generally, our finding illustrates that the transcriptional landscape of macrophages during a sustained exposure to a signal, even in a course of a few hours, undergoes a significant remodeling and a secondary stimulus is likely to elicit variable responses

depending on the exact timing of stimulation. Furthermore, given intrinsic macrophage plasticity, whereby a 12 hr treatment with a relevant signal (e.g. LPS or Dex) is sufficient to 'polarize' them to a distinct myeloid cell population – caution needs to be taken in interpreting results of 'sequential' treatments, which may document a response of a reprogrammed macrophage to a new signal rather that simple transcriptional antagonism or synergy.

Under conditions of GC repression, we observed a broad failure of paused genes to release NELF concomitantly with inhibition of Pol 2 entry into productive elongation. Moreover, genetic disruption of NELF resulted in GC resistance of genes in this class establishing a causal relationship between NELF accumulation and GR-mediated repression. Interestingly, NELF was previously shown to participate in estrogen receptor (ER) alpha-mediated gene expression. ERa primarily affects Pol 2 post-initiation steps, whereby pausing is alleviated via hormone-induced recruitment of CDK9 to Pol 2 and NELF and their phosphorylation (*Kininis et al., 2009*). Given that NRs can dynamically affect P-TEFb occupancy and that P-TEFb recruitment to GC-sensitive genes is attenuated after GC treatment in this and earlier studies (*Luecke and Yamamoto, 2005*; *Gupte et al., 2013*), GR may block elongation by preventing P-TEFb recruitment, possibly through direct steric hindrance. Interestingly, in addition to phosphorylation, NELF has recently been shown to undergo ADP-ribosylation which also facilitates its release (*Gibson et al., 2016*). It would be informative to assess whether, similar to P-TEFb, ADP-ribosyl transferases that modify NELF are susceptible to regulation by GCs. Finally, a physical interaction between ERa and NELF has been documented at promoters of certain estrogen-activated genes, where NELF recruitment limits the response to hormone (*Aiyar et al., 2004*). Conceivably, NELF could also serve as a non-conventional 'co-repressor' recruited by GR to the promoter-proximal regions of pro-inflammatory genes in a gene-specific manner. Once recruited, NELF may no longer require GR and assume its known function in Pol 2 pausing. Whether GR-mediated repression involves either of these mechanisms remains to be elucidated.

Interestingly, non-paused genes, such as Il1a and Il1b, exhibit low CpG content, stable nucleosome assembly at promoters, low levels of H3K9/14Ac in the basal state and low TBP occupancy (*Ramirez-Carrozzi et al., 2009*). This suggests that histone acetylation marks are required for chromatin remodeling which may pose a major barrier to the recruitment of Pol 2 at these genes. We show that an increase in H4Ac at promoters and kB sites in response to LPS correlated with Pol 2 recruitment, and GC attenuated these effects, suggesting that GR may repress these genes by acting upon factors that 'write' and 'read' histone marks. Among many HATs that modify H3 and H4, p300 is recruited by p65 to the TSS and NF-kB sites and has been shown to acetylate histones that are then bound by BRD4 (*Huang et al., 2009*; *Brown et al., 2014*; *Nagarajan et al., 2014*; *Roe et al., 2015*). Conceivably, GR attenuates p300 loading by competing for a tethering site on p65 as has been previously documented for IRF3 (*Ogawa et al., 2005*). We cannot exclude the possibility that additional HATs, that is, GCN5, contribute to writing H3/H4Ac at our GC-sensitive pro-inflammatory genes.

Given its role as a histone binding protein that reportedly contributes to recruiting P-TEFb and couples the acetylation state at promoters and enhancers with Pol 2 elongation, a clear bias for LPS-induced novel sites of BRD4 recruitment and their sensitivity to Dex specifically at non-paused genes was unexpected. BRD4 binding at promoters broadly correlates with gene activation (*Nicodeme et al., 2010*; *Lovén et al., 2013*; *Brown et al., 2014*; *Kanno et al., 2014*). We now show that similar to I-BETs, GR inhibits, albeit indirectly, loading of BRD4 particularly at non-paused genes and, by exploiting their dependency on histone acetylation, disrupts interactions with Mediator, ultimately antagonizing Pol 2 recruitment and transcription initiation. Because this effect is far from uniform, and some p65/BRD4-bound LPS-induced enhancers are more sensitive to the effects of Dex than others, we speculate that a subset of p65-binding sites has greater functional consequences for gene activity. Identifying a subpopulation of 'dominant' enhancers whose BRD4 occupancy is a definitive predictor of transcriptional state, and correlating those with sites of GR recruitment would likely sharpen the differences in BRD4 behavior between the two gene classes.

Finally, although the two classes of genes are activated and repressed through distinct mechanisms, the consequences of GR activation share commonalities including a failure to recruit P-TEFb and the Mediator complex. P-TEFb is required for gene activation post Pol 2 loading, so at non-paused genes failing to recruit Pol 2, P-TEFb loss would have little functional consequences. Conversely, a block in Mediator recruitment at both the TSS and kB sites could potentially contribute to repression of both classes of genes. Mediator is a multi-subunit complex that interacts with

numerous activators and components of basal transcription machinery including Pol 2 (*Malik and Roeder, 2010*). With respect to non-paused genes, Mediator interacts directly with both BRD4 and p300, with Mediator and BRD4 stabilizing each other's occupancy (*Jang et al., 2005*; *Malik and Roeder, 2010*; *Shi and Vakoc, 2014*). Furthermore, Mediator and p300 can act cooperatively to alter the chromatin landscape and facilitate PIC formation (*Malik and Roeder, 2010*). Although the contribution of Mediator to activation of pro-inflammatory paused genes needs further study, it has been suggested that Mediator may help recruit P-TEFb indirectly promoting pause release (*Lu et al., 2016*). Additionally, because kB sites are typically distant from promoters, and pro-inflammatory genes were proposed to be activated through DNA looping (*Tong et al., 2016*), Mediator (perhaps together with Brd4) may contribute to bridging promoters with NF-kB enhancers. Thus, it is tempting to speculate that by hindering Mediator assembly, GR globally disrupts promoter-enhancer communication thereby attenuating pro-inflammatory gene expression.

## Materials and methods

### Cell culture and reagents
BMDM were prepared from 8-to-10 week old mice as in *Gupte et al. (2013)*. RAW264.7 cells were cultured in DMEM media (Corning, cat# 10–013-CV) supplemented with 10% fetal bovine serum (Atlanta Biologicals cat# S11550). Dex and LPS were purchased from Sigma.

### Transgenic mice
C57BL/6 mice (NCI, Charles River Laboratories), C57BL/6 *LysM-Cre* mice -/-:*Nelfb* fl/fl mice and their derivatives were maintained in the Weill Cornell Animal Facility in compliance with guidelines from the Weill Cornell Animal Care and Use Committee. 8-to-10-week-old male mice were used for bone marrow isolation.

To create the NELF-B conditional KO strain, *Nelfb* fl/fl mice (with *Nelfb* exon 4 floxed [*Amleh et al., 2009*]) were bred to C57BL/6-derived *LysM-Cre* mice (Jackson Laboratories, 004781) to obtain double heterozygous *LysM-Cre*/wt:*Nelb* fl/wt (*LysM-Cre*-HET) animals. To create homozygous (*LysM-Cre*:*Nelfb* fl/fl) animals, we self-crossed *LysM-Cre*-HET mice. The genotype of the progeny was determined using PCR primers described in *Amleh et al. (2009)*. *LysM-Cre* primers were obtained from Jackson Laboratories.

### Inhibitor experiments
BMDM were plated in 6-well plates at $2*10^6$ cells/well. For BRD inhibitor experiments, cells were pretreated with I-BET (Calbiochem, 401010) for 30 min, followed by co-treatment with LPS (10 ng/ml). For p300 inhibitor experiments, cells were treated with LPS for 30 min, followed by co-treatment with C646 (Abcam, ab142163) for 1 hr. Concentrations of inhibitors are shown in Figure Legends.

### Transfections
RAW264.7 cells were plated at $2*10^5$ cells/well in 24-well plates and transfected ON using Turbofect (Thermo Scientific, R0531) as per manufacturer's instructions. Cells were treated the following day as described in Figure Legends. Plasmids used are pcDNA3.1-p300, pcDNA3.1-300(HAT-) (Addgene, Plasmid #23252 and #23254, respectively) and pcDNA3.1 to equalize total amount of transfected DNA.

### RNA isolation and real-time qPCR
Total RNA isolation from BMDM (Qiagen RNAeasy Kit), random-primed cDNA synthesis, and qPCR with Maxima Sybr Green/ROX/2x master mix (Fermentas) on StepOne Plus real time PCR system were performed using standard protocols. Data analysis was performed using the ddCT method. All data were normalized to *Actb* as housekeeping control. Primers are listed in *Supplemental file 4*.

### Immunoblotting
Whole cell extracts were prepared in RIPA buffer (10 mM Tris-HCl pH 8.0, 1 mM EDTA, 0.5 mM EGTA, 140 mM NaCl, 5% glycerol, 0.1% Na deoxycholate, 0.1% SDS, 1% Triton X-100).

Immunoblotting was performed with rabbit polyclonal antibodies to NELF-B (Cell Signaling, 1:2000, 1489S), NELF-E (Proteintech, 1:2000, 10705–1-AP), HSP90 (Cell Signaling 1:200, 4874S).

## ChIP

BMDM were treated for 45 min as specified in Figure Legends and single cross-linked in 1% methanol-free formaldehyde for 10 min at RT (AcH4) or double cross-linked using 2 mM disuccinimidyl glutarate (Proteochem, c1104) for 30 min followed by 1% methanol-free formaldehyde for 10 min at RT (CDK9, BRD4, MED1, MED12, p300). The reaction was quenched by 0.125 M glycine for 5 min. Cells were then washed with PBS, scraped and lysed for 10 min at 4°C in lysis buffer with protease inhibitor cocktail. The nuclear extracts were collected by centrifugation at 600*g for 10 min. Nuclei were then washed for 10 min at 4°C in wash buffer with protease inhibitors and collected as described above. Nuclei were lysed in lysis buffer for 10 min and sonicated to fragment chromatin using 15–18 cycles (30 s 'on', 30 s 'off') in a Bioruptor at 4°C. For CDK9, nuclei were sonicated with Covaris S220 Ultrasonicator according to manufacturer's instructions (130 µl shearing buffer, 200 cycles/burst, 120 s, DF 10). Lysates were cleared by centrifugation at 14,000*g, 20 min, 4°C, and then incubated with normal rabbit IgG (Santa Cruz Biotech, sc-2027x), BRD4 (Abcam, ab84776 and Bethyl Laboratories, A3001-985A100), MED12 (Bethyl Laboratories, A300-774A), MED1 (Bethyl Laboratories, A300-793), p300 (Santa Cruz Biotech, sc-585X), Anti-AcH4 (Millipore, 06–866), Anti-AcH4K12 (Millipore, 07–595), Anti-AcH4K5 (Millipore, 07–327) and 40 µl of 50% protein A/G plus agarose (Santa Cruz Biotech, sc-2003) per reaction at 4°C ON. Beads were washed 4x with RIPA buffer and once with TE buffer. For CDK9, 5 µg of antibodies (Santa Cruz Biotech, sc-8338X or sc-13130X) were pre-bound to 40 µl of Dynabeads Protein A (Invitrogen), washed 2x with beads blocking buffer and incubated with lysates at 4°C ON; IPs were washed 6x with modified RIPA buffer containing 100 mM LiCl on a magnetic stand and once with TE buffer +50 mM NaCl.Each reaction was then incubated in TE + 0.5% SDS +200 µg/ml proteinase K (Invitrogen, 25530049) for 2 hr at 55°C, followed by 6 hr at 65°C to reverse crosslinks. DNA was purified using phenol-chloroform extraction and ethanol precipitation or using Qiagen PCR purification kit. Recruitment at binding sites was assessed by qPCR. All data at putative binding sites were normalized to 28S ribosomal RNA as control. Primers are listed in *Supplemental file 4*.

## ChIP-seq

For GR (Santa Cruz Biotech, sc-1004X), BRD4 (Abcam, ab84776) and p65 (Santa Cruz Biotech, sc-372X) ChIP-seq, nuclei were sonicated with Covaris S220 sonicator according to manufacturer's instructions (130 µl shearing buffer, 200 cycles/burst, 120 s, DF 10). For Pol 2 (Santa Cruz Biotech, sc-9001X) and NELF-E (Proteintech, 10705–1-AP) ChIP-seq, cells were formaldehyde cross-linked and nuclei were sonicated as above to obtain fragments in 150–500 bp range. Input DNA was prepared from sonicated material saved prior to IP. Lysates were cleared by centrifugation at 14,000 rpm, 20 min, 4°C, and then incubated with respective antibodies using 40 µl of 50% protein A/G PLUS agarose beads (for GR, BRD4 and Pol 2) or 60 µl of Dynabeads (Invitrogen) (for p65) per reaction at 4°C ON. GR, BRD4 and Pol 2 IPs were then processed as described for ChIP-qPCR above. p65 IPs were washed 6x with modified RIPA buffer containing 100 mM LiCl on a magnetic stand and once with TE buffer +50 mM NaCl and processed as described for ChIP-qPCR above. The efficiency of ChIP was assessed by qPCR. The integrity and quality of DNA was evaluated with Bionalyzer 2100 (Agilent Technologies) before using 10 ng of material to prepare Illumina-compatible sequencing libraries using Illumina Truseq ChIP sample prep kit. Library preparation and sequencing was performed by Weill Cornell Epigenomics Core. Libraries were sequenced by a HiSeq 2500 (50 bp, single-end).

## RNA-seq

BMDM from LysM-Cre:NELF-B wt/wt (WT) and/or LysM-Cre:NELF-B fl/fl (NELF-B KO) mice were treated as indicated in individual figure legends (vehicle, LPS, LPS + Dex for 1 hr) and RNA was isolated using Qiagen RNA-easy kit. Total RNA was polyA enriched and converted into Illumina-compatible sequencing library with TruSeq mRNA-Seq sample preparation kit (Illumina). Quality control of RNA and libraries was performed using the BioAnalyzer 2100. Pair-end sequencing was performed at the Weill Cornell Epigenomics Core using HiSeq2500.

## Quantification and statistical analysis
### General experimental design and statistical analysis
To ensure reproducibility all in vitro experiments were repeated at least in triplicates. The differences between continuous variables were assessed using Student's *t* Test and the differences between discrete variable were assessed with Fisher's exact test.

### Real time PCR
Two-tailed Student's t-test was used to ascertain the differences between means as detailed in Figure Legends.

### ChIP-seq
Sequencing quality control was performed using FASTQC; adapters, when needed, where trimmed using trimmomatic. 50 bp single-end reads were aligned to the mouse genome (mm10) using CLC Bio Genomic Workbench (GR, Pol 2) or bowtie2 (p65, NELF-E, BRD4). Aligned BAM files were converted into bigwig format for data visualization purposes. The quality of Chip-seq experiments was assessed using ChIPQC package (*Carroll et al., 2014*) (*Supplemental file 2*). Cross-correlation analysis, Relative Strand Correlation (RSC) and Normalized Strand Cross-correlation coefficient (NSC) for all ChIP-seq datasets used in this study were calculated with CLC BIO genomics workbench (*Figure 1—figure supplements 1D* and *2C*; *Figure 3—figure supplement 1A* and *Figure 4—figure supplement 1C*, *Supplemental file 2*) as described in *Marinov et al., 2014*. RSC reflects the ratio of the fragment-size peaks and the read-size peak in cross-correlation plot. For all experiments with the exception of one NELF-E condition, the RSC is larger than 0.8 as per ENCODE recommendations (*Landt et al., 2012*). Peak calling was performed with CLC Bio Genomics Workbench (Pol 2) or MACS2 (*Zhang et al., 2008*) (–gsize 2150570000 –bw=300 ratio 1.0 –slocal 1000 –llocal 10000 –keep-dup 1 –bdg –qvalue 0.05) with a matching input file to estimate background read distribution.

Peak annotation relative to known genomics features was performed using *ChIPpeakAnno package (R, Bioconductor)* (*Zhu et al., 2010*) with *TxDb.Mmusculus.UCSC.mm10.knownGene* annotation (2016-09-29 04:05:09 + 000). Peak overlaps between datasets were determined using *subsetByOverlap* function from GenomicRanges package (R, Bioconductor) with the minimum overlap of 1 nt and visualized with *makeVennDiagram* function from *ChIPpeakAnno* (*Zhu et al., 2010*) package.

*Ab initio* analysis of overrepresented sequences in ChIP-seq peaks was performed using MEME-ChIP suite with MEME (long sequences), DREME (short sequences) and CentriMO (centrally-enriched sequences). E-values estimate the expected number of motifs in an experimental set of sequences compared to random sequences of the similar size. Sequencing motifs with E-values under 0.0001 were considered statistically significant.

Pol 2 pausing indexes (PI) were calculated as previously described (*Nechaev et al., 2010*). All transcripts for LPS-induced Dex-sensitive genes present in *TxDb.Mmusculus.UCSC. mm10.knownGene* annotation were filtered to collapse all annotated transcripts with identical 5' ends to a single gene model. For remaining transcripts, the PI was calculated as the ratio of log-transformed, length-normalized read counts at the 5' end flanking region ($-200:+500$) and transcript 'body' ($+500$: end of a transcript). To compare between replicates, the PI were normalized to respective library sizes (as in *Figure 2B*). Read distributions in the region of interest ('promoters' and gene 'bodies') were visualized in the form of 'heat' maps that show scores (coverage) at a given sequence position or bin using *genomation* package (R, Bioconductor) (*Akalin et al., 2015*). For heat maps visualization, paused and non-paused transcripts were further filtered by selecting only those that had overlapping Pol 2 peaks in the 'promoter' area in LPS-treated BMDM. To summarize read distributions, we plotted mean coverages (*plotMeta, genomation*) over regions of interest (*Figures 2D* and *3A* and *Figure 4—figure supplement 1B*) with the standard error and the 95% confidence interval bands.

### RNA-seq
RNA-seq analysis has been performed as previously described (*Coppo et al., 2016*). 50 bp paired reads were mapped to annotated mouse genome (mm10) with CLC Bio Genomic Workbench (Qiagen).Read count table containing unique exon reads was analyzed using EdgeR (*Robinson et al., 2010*) package to determine differentially expressed genes. Read counts were scale normalized using the weighted trimmed mean method and expressed as log-transformed counts per million

(cpm). All genes with unadjusted p-value<0.01 (p<0.05 for NELF-B KO experiment) and fold change >1.5 in at least one pairwise comparison were considered to be differentially expressed and were selected for further analysis.

## Accession numbers

All raw sequence data generated in this study are deposited to NCBI GEO: GSE110279 https://www.ncbi.nlm.nih.gov/geo/query/acc.cgi?acc=GSE110279

## Acknowledgements

We are grateful to CE Mason and JA Gandara (Weill Cornell) and S Mimouna (HSS) for technical help. We acknowledge help from the A Alonso, Y Li and the staff of Weill Cornell Epigenomics core. We thank S Mimouna for helpful discussion. MAS is supported by the NIH Diversity Supplement 3R01DK099087-01A1S1 and the MSTP grant T32GM007739 from the NIH NIGMS to the Weill Cornell/ Rockefeller/ Sloan-Kettering Tri-Institutional MD-PhD Program. This work is supported by the grants to IR from NIH R01DK099087, the Rheumatology Research Foundation Research Grant, the DOD CDMRP PR130049 Research Award and The Hospital for Special Surgery David Rosensweig Genomics Center. XH is supported by the Ministry of Science and Technology of China National Key Research Project 2015CB943201, National Natural Science Foundation grants 81422019, 81571580, 91642115, 8151101184, and funds from Tsinghua-Peking Center for Life Sciences. RL is supported by the NIH R01CA220578.

## Additional information

### Funding

| Funder | Grant reference number | Author |
|---|---|---|
| National Institutes of Health | R01DK099087 | Maria A Sacta<br>Bowranigan Tharmalingam<br>Maddalena Coppo<br>David A Rollins<br>Dinesh K Deochand<br>Bradley Benjamin<br>Yurii Chinenov<br>Inez Rogatsky |
| U.S. Department of Defense | PR130049 | Bowranigan Tharmalingam<br>Maddalena Coppo<br>Yurii Chinenov<br>Inez Rogatsky |
| Rheumatology Research Foundation | | David A Rollins<br>Yurii Chinenov<br>Inez Rogatsky |
| Hospital for Special Surgery David Rosensweig Genomic Center | | Maddalena Coppo<br>Yurii Chinenov<br>Inez Rogatsky |
| Ministry of Science and Technology of the People's Republic of China | | Li Yu<br>Bin Zhang<br>Xiaoyu Hu |
| National Natural Science Foundation of China | 81422019 | Li Yu<br>Bin Zhang<br>Xiaoyu Hu |
| Tsinghua University | | Li Yu<br>Bin Zhang<br>Xiaoyu Hu |
| National Institutes of Health | R01 CA220578 | Rong Li |
| National Natural Science Foundation of China | 81571580 | Li Yu<br>Bin Zhang<br>Xiaoyu Hu |

| National Natural Science Foundation of China | 91642115 | Li Yu Bin Zhang Xiaoyu Hu |
| National Natural Science Foundation of China | 8151101184 | Li Yu Bin Zhang Xiaoyu Hu |
| National Institutes of Health | T32GM007739 | Maria A Sacta |

The funders had no role in study design, data collection and interpretation, or the decision to submit the work for publication.

## Author contributions

Maria A Sacta, Conceptualization, Formal analysis, Supervision, Validation, Investigation, Visualization, Methodology, Writing—original draft; Bowranigan Tharmalingam, Formal analysis, Validation, Investigation; Maddalena Coppo, Data curation, Investigation, Methodology; David A Rollins, Data curation, Validation, Investigation, Methodology; Dinesh K Deochand, Formal analysis, Investigation, Methodology; Bradley Benjamin, Formal analysis, Validation, Investigation, Methodology; Li Yu, Investigation, Methodology; Bin Zhang, Software, Formal analysis; Xiaoyu Hu, Conceptualization, Formal analysis, Supervision, Funding acquisition; Rong Li, Resources; Yurii Chinenov, Software, Formal analysis, Validation, Investigation, Visualization, Writing—original draft, Writing—review and editing; Inez Rogatsky, Conceptualization, Data curation, Formal analysis, Supervision, Funding acquisition, Investigation, Writing—original draft, Project administration, Writing—review and editing

## Author ORCIDs

Bin Zhang http://orcid.org/0000-0001-6232-6768
Rong Li http://orcid.org/0000-0002-6471-6580
Inez Rogatsky http://orcid.org/0000-0003-3514-5077

## Ethics

Animal experimentation: This study was performed in strict accordance with the recommendations in the Guide for the Care and Use of Laboratory Animals of the National Institutes of Health. Mice were maintained in the Weill Cornell Animal Facility in compliance with guidelines from the Weill Cornell Animal Care and Use Committee (Protocol approval # 2015-0050).

## Decision letter and Author response

Decision letter https://doi.org/10.7554/eLife.34864.025
Author response https://doi.org/10.7554/eLife.34864.026

# Additional files

### Supplementary files

- Supplementary file 1. Summary of RNA-seq data for LPS-induced Dex-repressed genes.
DOI: https://doi.org/10.7554/eLife.34864.016

- Supplementary file 2. Summary of ChIP-seq experiments.
DOI: https://doi.org/10.7554/eLife.34864.017

- Supplementary file 3. Pausing Indexes for LPS-induced Dex-repressed transcripts.
DOI: https://doi.org/10.7554/eLife.34864.018

- Supplementary file 4. Primer pairs used in the study.
DOI: https://doi.org/10.7554/eLife.34864.019

- Transparent reporting form
DOI: https://doi.org/10.7554/eLife.34864.020

### Major datasets

The following dataset was generated:

| Author(s) | Year | Dataset title | Dataset URL | Database, license, and accessibility information |
|---|---|---|---|---|
| Sacta MA, Tharmalingam B, Coppo M, Rollins DA, Deochand DK, Benjamin B, Yu L, Zhang B, Hu X, Li R, Chinenov Y, Rogatsky I | 2018 | Gene-specific mechanisms direct Glucocorticoid Receptor-driven repression of inflammatory response genes in macrophages | https://www.ncbi.nlm.nih.gov/geo/query/acc.cgi?acc=GSE110279 | Publicly available at the NCBI Gene Expression Omnibus (accession no: GSE110279) |

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
