## [Decision Letter]

[Editors’ note: a previous version of this study was rejected after peer review, but the authors submitted for reconsideration. The first decision letter after peer review is shown below.]

Thank you for submitting your work entitled "Gene-specific mechanisms direct Glucocorticoid Receptor-driven repression of inflammatory response genes in macrophages" for consideration by *eLife*. Your article has been evaluated by a Senior Editor and three reviewers, one of whom is a member of our Board of Reviewing Editors. The following individual involved in review of your submission has agreed to reveal their identity: Iván D'Orso (Reviewer #3).

The main findings were thought to be of general interest, but after considerable discussion among the Senior Editor, Reviewing Editor and reviewers, it was concluded that the essential revisions needed to substantiate the major conclusions would require significantly more than a two month time frame. The decision is therefore to reject the manuscript, but to include the distilled 'Essential revisions' resulting from editorial discussion in addition to the full individual reviews. The reviewers indicated willingness to consider a revised manuscript that adequately addresses these main concerns should you wish to pursue that option.

Sacta et al. investigates the mechanisms of GC mediated repression on inflammatory genes. Using genome-wide approaches they identify the inflammatory gene set that is repressed by GCs. Based on PolII pausing index they identify inflammatory genes that are paused and become activated by the inflammatory stimulus by a mechanism involving the release of NELF. Genes that are not paused recruit PolII de novo, but both paused and not paused inflammatory gene groups are susceptible to GC mediated repression. GR seems to repress paused genes by depositing NELF, while at non-paused genes it inhibits the recruitment of p300, which will ultimately lead to a defect in BRD4 and Mediator assembly, diminishing transcription. Many prior studies have investigated GR repression, and there is substantial overlap in phenomenology reported here and the existing literature. The novelty relates to dividing GR repressed genes into those that are regulated at a pause-release step and those that are not. These findings are interesting, but there are a number of concerns about the strength of the main conclusions. These relate to reproducibility of genome-wide analysis, evidence for the tethering mechanism, relationship of GR binding to NELF occupancy, and interpretation of p300 over expression studies.

Essential revisions:

1) There do not appear to be replicates for ChIP-seq experiments. The authors use a cross correlation analysis method to estimate the quality of their ChIP-seq data, but this is not the same as evaluating reproducibility across experiments. The findings are probably sound with respect to peak locations, but the lack of replicates could be problematic when attempting to make quantitative comparisons and using arbitrary thresholds to divide features into one category or another. All three reviewers were of the opinion that replicates are required to make genome-wide conclusions, which are an essential aspect of this manuscript. The authors should provide a table indicating characteristics of each of the high throughput samples, including number replicates, number of uniquely mapped reads and for ChIP-seq experiments the fraction of reads in peaks. For replicates, the similarities of samples should be indicated e.g., by a Pearson correlation.

2) The authors identified 201 liganded GR-repressed inflammatory genes (Figure 1). They suggest that the dominant mechanism of GR-mediated repression is rapid tethering of GR to LPS-activated p65. However, they did not show whether these GR-repressed inflammatory genes are sensitive or insensitive to receptor activation in Dex treated macrophages without inflammatory stimuli. Sensitivity to Dex in the absence of LPS stimulation would suggest a mechanism independent of tethering by NF-κB.

3) The tethering mechanism itself needs to be further substantiated. At this point, the evidence is based on poor enrichment of GRE sequences at sites associated with binding of p65. There is substantial prior evidence that GR can interact with NF-κB. An important question is whether GR is interacting at negatively regulated sites by tethering or by binding to unconventional GREs, which have also been suggested as a basis for negative regulation. This question may be difficult to address in the BMDMs in a reasonable time frame, but could be addressed by generating by using CRISPR/Cas9 approaches to introduce DNA binding mutants in the endogenous GR locus in the RAW system and performing ChIP-seq experiments under each experimental condition. (An alternative would be to study GR in BMDMs in the GRdim mice, if available). These experiments would clearly establish which binding events were dependent on the DNA binding domain and which were mediated by tethering.

4) The findings regarding the requirement of NELF for GR repression of paused genes are interesting, but more information is needed for interpretation. Does GR treatment prevent dismissal of NELF from paused promoters? Is there a consistent relationship between GR occupancy at the promoters of paused genes and the importance of NELF? How does NELF affect basal gene expression at paused inflammatory genes and intergenic enhancers? If NELF keeps these genes under pausing, in the absence of NELF at least in theory gene expression should increase. It would be helpful to provide the absolute levels of expression of these genes rather than fold change to enable a more complete assessment of the consequences of NELF deletion.

5) Further related to the pause-release mechanism, a study by Zhu et al. (Biochemistry 2011) showed that the master regulator of Pol II pause release (the Cdk9/P-TEFb kinase) is a competitive decelerator of GR transactivation activity and does not interfere with the inhibitory activity of NELF. What is the role of CDK9 on NF-κB and GR binding in the repression of inflammatory genes? Along the same line, one key relevant article (PMID 15879558) was not discussed in the context of the discoveries of this manuscript. In that paper, Luecke and Yamamoto published that GR blocks recruitment of the P-TEFb kinase by NF-κB to effect promoter-specific transcriptional repression. Is there any relationship between these previous discoveries and the mechanisms of LPS+Dex stimulation/repression discussed in this manuscript?

6) An old model for nuclear receptor-dependent repression of inflammatory response genes posited re-distribution of coactivators upon nuclear receptor activation. Here, most GR sites are not at NF-κB peaks and could potentially compete for the binding of p300 for gene activation. This mechanism would also result in loss of p300 at p65 peaks and be overcome by overexpression of wild type p300 but not mutant p300. Therefore, the experiments presented in Figure 5 do not clearly demonstrate that local tethering of GR is the cause of reduced p300 and histone acetylation at these sites. Examination of p300 and histone acetylation, etc., in a cell engineered to express a DNA binding mutant that cannot bind to specific DNA sites (and thus recruit p300 to these sites) but can still tether to NF-κB sites, as requested in point 3, above, would be needed to address this concern.

*Reviewer #1:*

The manuscript from Sacta and colleagues reports on the analysis of dexamethasone (dex) mediated repression of genes upregulated upon LPS stimulus either by release of the paused RNA polymerase II or de-novo recruitment thereof. They find that both, control of RNAP II elongation or de-novo recruitment are equally susceptible to GR repression. The authors present evidence of GR-binding to p65 sites upon LPS treatment and reduced dex-induced repression of paused genes upon NELF depletion, a factor pivotal for Pol II pausing. In addition, they provide evidence for the opposition of p300, BRD4 and GR. The findings regarding two distinct mechanisms for GR repression are of general interest, but some of the main conclusions are not yet sufficiently established.

The findings regarding the requirement of NELF for GR repression of paused genes are interesting, but more information is needed for interpretation. Does GR treatment prevent dismissal of NELF from paused promoters? Is there a consistent relationship between GR occupancy at the promoters of paused genes and the importance of NELF? What happens to the basal expression of these genes? Are they still induced by LPS or do they become constitutively activated? It would be helpful to provide the absolute levels of expression of these genes rather than fold change to enable a more complete assessment of the consequences of NELF deletion.

For the RNAseq, a rational for the relatively relaxed conditions of an FDR to 0.1 and using a 1.5-fold change should be presented. How far did the additionally obtained genes improve the findings compared to the more commonly utilized i.e. FDR 0.05 and >2 fold? A more detailed description of the rationale and used parameters would benefit the methods.

The authors do not relate the present findings to their earlier studies indicating a requirement for GRIP1 in repression of LPS-dependent gene expression. Is there any relationship to GRIP-dependent repression and either class of genes (paused vs. non paused)?

An old model for nuclear receptor-dependent repression of inflammatory response genes posited re-distribution of coactivators upon nuclear receptor activation. Here, most GR sites are not at NF-κB peaks and could potentially compete with p300 for gene activation. This mechanism would also result in loss of p300 at p65 peaks and be overcome by overexpression of wild type p300 but not mutant p300. Therefore, the experiments presented in Figure 5 do not clearly demonstrate that local tethering of GR is the cause of reduced p300 and histone acetylation at these sites. Some alternative approach would be needed, e.g., a DNA binding mutant that cannot bind to specific DNA sites (and thus recruit p300 to these sites) but can still tether to NF-κB sites.

There do not appear to be replicates for ChIP-seq experiments. The authors use a cross correlation analysis method to estimate the quality of their ChIP-seq data, but this is not the same as evaluating reproducibility across experiments. The findings are probably sound with respect to peak locations, but the lack of replicates could be problematic when attempting to make quantitative comparisons and using somewhat arbitrary thresholds to divide features into one category or another. The authors should provide a table indicating characteristics of each of the high throughput samples, including number of uniquely mapped reads and for ChIP-seq experiments the fraction of reads in peaks. For replicates, the similarities of samples should be indicated e.g., by a Pearson correlation. In the absence of replicates, the key findings from ChIP-seq need to be confirmed at informative loci for representative genes.

The manuscript would benefit from a discussion of previous findings including the physical interaction of NF-κB and GR upon GR translocation (Ray laboratory and others, i.e. doi: 10.1073/pnas.91.2.752). Treatment with alternative ligands such as RU486 which facilitates GR translocation but negligible activity to inhibit NF-κB transactivation would strengthen the claim of GR tethering at the described sites. The novelty of the finding is at this point not entirely clear and could benefit from a more thorough discussion of previous publications, particularly those from the Simons lab (doi: 10.1074/jbc.M115.683722 and most notably 10.1074/jbc.M115.683722), in which Brd4, NELF etc. are implicated in GR activation.

*Reviewer #2:*

Sacta et al. investigates the mechanisms of GC mediated repression on inflammatory genes. Using genome-wide approaches they identify the inflammatory gene set that is repressed by GCs. Based on PolII pausing index they identify inflammatory genes that are paused and become activated by the inflammatory stimulus by a mechanism involving the release of NELF. Genes that are not paused recruit PolII de novo, but both paused and not paused inflammatory gene groups are susceptible to GC mediated repression. GR seems to repress paused genes by depositing NELF, while at non-paused genes it inhibits the recruitment of p300, which will ultimately lead to a defect in BRD4 and Mediator assembly, diminishing transcription.

The manuscript offers some new insights about the mechanisms of GC mediated repression of inflammatory genes, however the tethering mechanism proposed for GR recruitment at GC repressed sites is not convincing and experimentally not confirmed. In the absence of this the enthusiasm of this reviewer is rather low, because the issue of GR mediated transcriptional repression has been addressed by many papers and this represents only an incremental advance. In addition the reviewer feels that the first part of the Results section is way too lengthy/complicated and contains more figures (motif enrichment analysis) that are not necessarily helping the main finding of the paper. Also the reviewer did not find any information about the number of replicates for the ChIP-seq experiments and there is no accession number provided to access sequencing datasets. The authors should provide a summary table containing all the next generation sequencing datasets.

1) The authors identified 201 liganded GR-repressed inflammatory genes (Figure 1). They suggest that the dominant mechanism of GR-mediated repression is to rapid tethering of GR to LPS-activated p65. However, they did not show whether these GR-repressed inflammatory genes are sensitive or insensitive to receptor activation in Dex treated macrophages without inflammatory stimuli.

2) The authors suggested that 111 new GR peaks appeared in association with GR-repressed inflammatory genes in LPS+Dex treated macrophages (Figure 1). In addition, they presented some representative examples of these peak set on Figure 1. However, Dex treated sample is missing on this figure.

3) Figure 1 suggests that in the proximity of Dex repressed genes, GR peaks enrich the motif of the receptor conflicting with the proposed tethering mechanism. The authors should clarify this and provide experimental evidence if the repression predominantly takes place via tethering.

4) The read distribution plot-based visualization of different GR peak sets (on Figure 1) is also necessary in untreated, LPS, Dex and LPS+Dex-treated macrophages.

5) The authors should be more careful with the title of Figure 1. "GR represses LPS-induced genes via p65-assisted tethering". Figure 1 neither does establish p65 as the main factor implicated in this, nor does provide evidence for the tethering mechanism (just by looking at motif enrichments one cannot claim that tethering is the main mechanism because of the absence of the GR motif). How do the authors know the relative contribution of p65 or other inflammatory transcription factors (AP1) to this "tethering"? Also if this is tethering, due to the fact that GR indirectly binds to these regions, I would expect to see smaller read enrichments (peaks) compared to regular, directly DNA bound GR sites.

6) On Figure 1. it seems that the vast majority of the GR binding sites are formed also in the absence of LPS, suggesting that GR independently of LPS can act at these sites. How does Dex pre-treatment followed by LPS stimulation affect gene expression at these loci? How does Dex treatment alone affect inflammatory gene expression at the basal state?

7) On Figure 1, are these GR peak categories overlap with p65 binding? The authors should make a schematic representation about the annotation process for easy understanding.

8) The authors claim that p65 binding is not affected by Dex, however there is no genome-wide analysis to back this claim. If the authors determined the RPKM values for p65 peaks on the regions identified on Figure 1 (Dex-repressed distal intergenic regions or other regions in this group) under the conditions presented would clarify if p65 binding indeed did not change.

9) The authors showed that the promoter regions of "paused" inflammatory genes were associated NELFE binding in absence of inflammatory stimuli. They described, that NELFE binding was decreased at these promoters after 1 hour LPS treatment. They suggest, that this factor also plays central role in GR-mediated repression. However, they did not show the deep comparison of NELFE binding between the unstimulated, LPS, Dex and LPS+Dex-stimulated macrophages at the promoter regions of all "paused" inflammatory genes (using read distribution plot-based visualization) or at the selected genes.

10) The authors described that 4 selected LPS-induced genes were repressed in a NelfB-dependent manner by Dex. How many genes do show similar NelfB dependency from the identified 201 liganded GR-repressed inflammatory genes?

11) Is the p65 binding regulated at the "paused" and "non-paused" inflammatory genes-associated enhancers/promoters by Dex?

12) Are the histone acetylation and p300 binding are regulated at the "non-paused" inflammatory genes-associated promoters/enhancers in the absence of inflammatory stimuli by GR activation?

13) How does NELF affect basal gene expression at paused inflammatory genes and intergenic enhancers? If NELF keeps these genes under pausing, in the absence of NELF at least in theory gene expression should increase.

14) How the absence of NELF affects the recruitment of p65 and GR at paused genes? Is it possible that pause release can negatively affect the recruitment of p65 or GR upon co-stimulation?

15) How Dex treatment affects NELF occupancy? Is it possible that GR activation deposits more NELF to inflammatory genes/enhancers and quenches future LPS response?

*Reviewer #3:*

In this manuscript Sacta et al. have utilized genome-wide and cell-based approaches to define mechanisms of glucocorticoid receptor (GR)-mediated repression of inflammatory genes in response to Dex treatment. GR potently represses the synthesis of inflammatory genes but the underlying mechanisms remained poorly understood. The authors used mouse bone marrow-derived macrophages treated ex vivo with LPS (to trigger the inflammatory program) and LPS+Dex co-treatment (to induce GR-mediated repression) as a system to tackle this scientific problem. They first classified GR target genes as paused and non-paused using an RNA Pol II pausing index criteria. While paused genes are activated by promoting Pol II pause release through eviction of the negative elongation factor NELF, non-paused genes are regulated by de novo Pol II recruitment to target promoters. The data suggests that GR does not preferentially repress genes in one-class vs the other. In addition, given that most of the GR regulated genes contain statistically enriched NF-κB (p65 subunit) binding motifs (as inferred from ChIP-seq data), the authors proposed a model in which the dominant repressive mechanism for both cases involves rapid GR 'tethering' to NF-κB. However, although the genomics data is convincing there is no evidence for the proposed recruitment mechanism and then the manuscript moves away from this central point to further define the molecular bases of the mechanism of activation and repression at both classes of genes (role of NELF on GR repression, role of histone acetylation, BRD4/Mediator recruitment to promoters, and p300 recruitment/activity). Striking differences between paused and non-paused genes were observed. While paused genes contain high-level of acetylation, non-paused genes do not but show a significant increase after LPS treatment, effect that was attenuated by Dex. They then analyzed the genome-wide distribution of BRD4 by ChIP-seq, but this section is confusing, not very well elaborated or integrated with other sections in the manuscript.

Finally, using a pharmacological approach, the authors proposed a function for p300 in the LPS-mediated activation of non-paused genes but not of paused genes, and that GR represses non-paused genes by precluding p300 recruitment, histone acetylation, BRD4 recruitment and Pol II loading. The study seems an extension of several others that have examined the transactivation and trans-repression roles of GR. Nevertheless, it has several interesting findings on how GR mediates its repressive effects and provides further insight into the potential cross talk between NF-κB and GR in the regulation of transcription. However, the implications of these findings in the context of inflammation/innate immunity, and key genetic evidence for the 'tethering' mechanism, are lacking. While the manuscript has novel and interesting observations, several points require further clarification and experimental support as indicated below.

1) It would be critical (especially for the reader not familiar with the system used in this manuscript) to explain why the authors used co-treatment (LPS+Dex) rather than pre-treatment with Dex to exert transcriptional repression prior to the LPS shock. There has to be a biological explanation that is not apparent in the text.

2) The conclusion that Dex repression of all 'paused' genes is attenuated upon loss of NELF is expected if GR functions through NELF, but there is no direct evidence supporting the model.

3) The final conclusion of section 1 arguing that NF-κB is a critical component of repression complexes is interesting but the model has not been tested. This is inexplicable. Previous studies have shown that GR interacts directly with the p65 subunit of NF-κB. Thus, do GR and NF-κB associate in order for GR to elicit its repressive effects? If so, it is unclear in the manuscript and must be highlighted. Then the manuscript transitions to an interesting topic (the role of NELF in GR repressive effects) but unrelated to the initial observations. This rough transition makes to think that the two parts were forced.

4) A study by Zhu et al. (Biochemistry 2011) showed that the master regulator of Pol II pause release (the Cdk9/P-TEFb kinase) is a competitive decelerator of GR transactivation activity and does not interfere with the inhibitory activity of NELF. What is the role of CDK9 on NF-κB and GR binding in the repression of inflammatory genes? Along the same line, one key relevant article (PMID 15879558) was not discussed in the context of the discoveries of this manuscript. In that paper, Luecke and Yamamoto published that GR blocks recruitment of the P-TEFb kinase by NF-κB to effect promoter-specific transcriptional repression. Is there any relationship between these previous discoveries and the mechanisms of LPS+Dex stimulation/repression discussed in this manuscript?

---

## [Author Response]

[Editors’ note: the author responses to the first round of peer review follow.]

Essential revisions:1) There do not appear to be replicates for ChIP-seq experiments. The authors use a cross correlation analysis method to estimate the quality of their ChIP-seq data, but this is not the same as evaluating reproducibility across experiments. The findings are probably sound with respect to peak locations, but the lack of replicates could be problematic when attempting to make quantitative comparisons and using arbitrary thresholds to divide features into one category or another. All three reviewers were of the opinion that replicates are required to make genome-wide conclusions, which are an essential aspect of this manuscript. The authors should provide a table indicating characteristics of each of the high throughput samples, including number replicates, number of uniquely mapped reads and for ChIP-seq experiments the fraction of reads in peaks. For replicates, the similarities of samples should be indicated e.g., by a Pearson correlation.

Reproducibility is certainly important, and we would like to clarify what constitutes a single ChIP-seq run for us when dealing with primary BMDM. A typical single 4-condition ChIP experiment requires bone marrow isolated from 4-6 mice. To then obtain enough sonicated ChIP material in 150-250 nt DNA fragment range for a sequencing experiment, we have to pool 3 to 8 (for particularly difficult cofactors or tethered GR) individual ChIPs (biological replicates), all of which are quality-controlled and have confirmed enrichments in expected locations by qPCR. This means, we need macrophages derived from 15 to 40 mice for a single 3-4-condition sequencing run. If we face a misfortune of the working antibody being discontinued (or Santa Cruz Biotech de facto going out of business), we have to start optimizations from scratch. This is what we have been doing over the past 5 months.

Our original manuscript contained replicate ChIP-seq runs for Pol 2. The revision contains 2 new 4-condition replicates for GR, a new 3-condition replicate for p65, and a new 3-condition replicate for Brd4. New data are integrated into revised Figure 1, Figure 1—figure supplement 1–Figure 1—figure supplement 2, Figure 4D-E, Figure 4—figure supplement 1. The requested similarities of replicates for GR, p65, Pol 2 and Brd4 are specifically shown in Figure 1—figure supplement 1 and Figure 1—figure supplement 2, Figure 3—figure supplement 1 and Figure 4—figure supplement 1, respectively. We include new Supplementary file 2that specifies additional ChIP-seq quality control metrics, as requested; the last column of this table details the number of individual biological replicate ChIPs that were pooled for each ChIP-seq run because we believe this information enables one to gauge the scale of these experiments.

At this point, the only unreplicated ChIP-seq is for NELF-E, because the original antibody is also no longer available . So we would have to re-optimize our ChIP protocol to yet another new antibody and perform both new ChIP-seq repeats (~8 individual 4-condition ChIPs). While this is doable, it would require at least 2 more months and, given how many additional experiments and at what cost we have completed (combined with the fact that thus far, the repeats have not challenged any of the original conclusions), we hope our reviewers will accept the NELF-E runs (composed of 2-3 experiments each) as they are.

2) The authors identified 201 liganded GR-repressed inflammatory genes (Figure 1). They suggest that the dominant mechanism of GR-mediated repression is rapid tethering of GR to LPS-activated p65. However, they did not show whether these GR-repressed inflammatory genes are sensitive or insensitive to receptor activation in Dex treated macrophages without inflammatory stimuli. Sensitivity to Dex in the absence of LPS stimulation would suggest a mechanism independent of tethering by NF-κB.3) The tethering mechanism itself needs to be further substantiated. At this point, the evidence is based on poor enrichment of GRE sequences at sites associated with binding of p65. There is substantial prior evidence that GR can interact with NF-κB. An important question is whether GR is interacting at negatively regulated sites by tethering or by binding to unconventional GREs, which have also been suggested as a basis for negative regulation. This question may be difficult to address in the BMDMs in a reasonable time frame, but could be addressed by generating by using CRISPR/Cas9 approaches to introduce DNA binding mutants in the endogenous GR locus in the RAW system and performing ChIP-seq experiments under each experimental condition. (An alternative would be to study GR in BMDMs in the GRdim mice, if available). These experiments would clearly establish which binding events were dependent on the DNA binding domain and which were mediated by tethering.

These two points are related, therefore, we address them below together.

We agree that indeed, binding to NF-κB sites in the absence of LPS stimulation (and hence NF-κB binding) would imply an alternative mechanism for GR recruitment. It should be noted, that a number of inflammatory genes are co-regulated by NF-κB and AP1, and certain level of basal/uninduced cJun binding seen in most cell types can potentially provide tethering surface to GR in the absence of additional stimulation. Importantly, inflammatory signaling is known to antagonize GR DNA binding and gene regulation due to both transcriptional and signaling mechanisms (see, for example, Rogatsky et al., PNAS 1998; Chinenov et al., BMC Genomics 2014). Thus, if GR represses our genes through direct DNA binding, it is expected to do it MUCH better in a Dex-alone condition.

RNA-seq experiments reported here did not include Dex-alone condition, however, our published BMDM RNA-seq analysis (Chinenov et al., PNAS 2012; Chinenov et al., BMC Genomics 2014) did include untreated and 1-h Dex-alone condition. We analyzed the data for our 201 genes and found that only 56 were repressed (1.3-fold threshold), and only 16 were repressed ≥2-fold. These data strongly argue for the necessity of inflammatory signal and LPS-induced p65 activation for recruiting GR, and certainly, against competitive binding. An additional column specifying the ability of GR to repress our 201 genes in the presence of Dex alone has been added to the revised Supplementary file 1.

Next, we re-analyzed motif enrichment given new datasets incorporated into the study. As detailed in Results, the striking part of the analysis is that within a new class of GR binding sites gained specifically in response to LPS (LPS+Dex – unique sites), relative to Dex – unique or Dex: LPS+Dex – overlapping, is the NF-κB motifs, whereas GREs are not enriched. In contrast, the GR Dex – unique or Dex: LPS+Dex – overlapping peaks reveal the GREs and lack NF-κB enrichment (revised Figure 1). We now also added Dex-alone GR ChIP-seq read density distributions in revised Figure 1(Tnf, Il1b, Il12b, Cd83): new tracks demonstrate that GR binding peaks appear ONLY in LPS+Dex condition and perfectly overlap p65 peaks; no GR binding is seen in Dex-alone-treated BMDM. We see a similar LPS+Dex – specific GR binding by regular ChIP in both BMDM and human macrophage-like THP1 cells (Rollins et al., Nature Commun 2017 – Figure 6D-E https://www.nature.com/articles/s41467-01701569-2). Additional sites of GR and p65 co-binding at NF-κB enhancers in LPS+Dex-cotreated BMDM are shown in revised Figure 4. Thus, binding data are consistent with the failure of the majority of these genes to be repressed by GR in the absence of inflammatory stimulation and NF-κB binding.

Of course, using ‘uncoupling’ mutants has been attempted many times over 25 years since tethering has been first described, but turned out to be less straightforward than one would wish. The X-ray structures of GR bound to NF-κB are not as yet available, but GR DBD is well documented to be required for this interaction (see, for example, Nissen and Yamamoto, Genes and Dev. 2000); many mutations that affect GR DNA binding also affect these protein:protein interactions as well as GR DNA binding specificity. Using the infamous GR Dim mutant (and Dim-expressing BMDM), for example, is not a viable strategy as the mutant has been historically misrepresented as ‘DNA binding-deficient’ and even ‘activation-deficient’, when later work revealed that it is a complex gain-of-function transcription factor with altered DNA binding specificity, perfectly able to still bind to and regulate a subset of WT GR targets as well as NEW genes not regulated by the WT GR (Schiller et al., Genome Biology 2014). In addition, RAW cells brought up in reviews are extremely genetically unstable; every lab utilizes their favorite ‘strain’ of RAW cells which bear little resemblance to gene expression patterns of the primary macrophages and rapidly evolve in culture over as little as a few passages precluding a ‘stable line selection’ approach.

Upon consulting with the editors, we decided to introduce tagged versions of WT vs. DNA-binding-deficient GR of choice into immortalized (i) BMDM and to examine their recruitment to a few induced vs. repressed genes by ChIP. It was agreed that we would not be able to analyze activities of these mutants given the presence of endogenous GR in iBMDM, but occupancy should be sufficient in light of extensive evidence for tethering described above.

We created HA-tagged versions of WT hGR and two DNA-binding domain mutants C421G and K442E (https://www.ncbi.nlm.nih.gov/pubmed/3191531; https://www.ncbi.nlm.nih.gov/pubmed/16239257) in which the 1st Zn finger and the 1st helix of the DNA recognition motif, respectively, were disrupted. We packaged these GR derivatives into lenti-viruses using HEK293T cell-packaging line and transduced iBMDM to select for expressors by FACS based on the dsRed2 fluorescent protein-encoding gene in the vector (EF.CMV.RFP https://www.ncbi.nlm.nih.gov/pubmed/12788657 – Addgene). We discovered through multiple attempts, however, that iBMDM shut down ectopic GR expression while retaining dsRed2 fluorescence. As an alternative strategy then, we switched to GR-negative U2OS cells that would at least enable us to assess both activity and binding of GR proteins in the same context. We recognize that this experiment in non-immune, human cells with transiently transfected GR is quite a departure from the primary BMDM work which remains the central goal of our study, and therefore present as supporting ‘proof-of-principle’ data for the reviewers in this rebuttal (Author response image 1).

**Author response image 1. respfig1:** GR DNA-binding mutants are competent for repression at NF-κB GR-tethering sites. U2OS cells were transfected with hGR (WT, C421G or K442E)-expressing or empty EF.CMV.RFP vector (a, left) using Lipofectamine 3000 (ThermoFisher) and treated the next day with vehicle (Veh), 100 nM Dex, 5 ng/ml TNF or TNF+Dex, as indicated. (**a**) Gene expression of GR-activated IGFBP1 and GILZ (middle) or GR-repressed TNF and IL8 (right) following 2 h treatment was assessed by RT-qPCR with b-Actin as housekeeping control. IGFBP1 (n=5) and GILZ (n=4) fold induction by Dex is expressed over transcript level in veh-treated cells (=1). TNF and IL8 expression (n=3 each) is calculated relative to the transcript level in TNF-treated cells (=1). (**b**) ChIP with polyclonal antiGR Abs (PA1-511A ThermoFisher) was performed following 2-h Dex (left) or 1-h TNF or TNF+Dex (right) treatment, and GR occupancy was assessed at GREs and NF-κB sites at indicated genomic locations with signals at 28S ribosomal gene as normalization control (n=3). Occupancy is expressed over baseline in veh-treated cells (=1). Error bars are SEM; 2-tailed Student’s t-test.

As expected, both mutants display severely compromised ability to activate known GR target genes IGFBP1 and GILZ that are induced by the WT GR in Dex-treated U2OS cells (Author response image 1, left and middle). In contrast, the NF-κB target genes TNF and IL8 that are activated by TNF (TNF, a physiological inducer of NF-κB activity, is used instead of LPS because U2OS cells are not very LPS-responsive due to poor expression of TLR4) were repressed similarly by the WT GR and its DNA-binding mutants (Author response image 1, right). Upon scaling up transfection experiments for ChIP, we were able to detect WT GR, but not DNA-binding mutants, at the two GREs of IGFBP1 and at GILZ after 2-h Dex exposure (Author response image 1, left). In contrast, all three GR derivatives occupied NF-κB sites of TNF and IL8 upon 1-h TNF+Dex treatment (Author response image 1, right), further corroborating a tethering model in which p65 is a critical component of GR repression complexes, even in this heterologous experimental system.

4) The findings regarding the requirement of NELF for GR repression of paused genes are interesting, but more information is needed for interpretation. Does GR treatment prevent dismissal of NELF from paused promoters? Is there a consistent relationship between GR occupancy at the promoters of paused genes and the importance of NELF? How does NELF affect basal gene expression at paused inflammatory genes and intergenic enhancers? If NELF keeps these genes under pausing, in the absence of NELF at least in theory gene expression should increase. It would be helpful to provide the absolute levels of expression of these genes rather than fold change to enable a more complete assessment of the consequences of NELF deletion.

We apologize if we did not make this clear. Indeed, failure to dismiss NELF is a key component of GR repression of genes in the paused class. Genome-wide NELF occupancy in LPS+Dex (repressed) BMDM is indistinguishable from that seen in resting macrophages, so ‘paused’ configuration (which is rapidly disrupted when NELF is released in response to LPS) instead persists in the presence of LPS+Dex (Figure 3).

We agree that a priori, one may expect that the loss of NELF (NELF-B KO) would release paused Pol II leading to ‘leaky’ expression; however, an equally plausible scenario is that NELF ‘concentrates’ initiated transcriptionally engaged Pol II at the TSS, in which case its loss leads to the loss of Pol II and the baselines would be unaffected or even drop in NELF-B KO. In our system, we see no broad effect of NELF deletion on our genes of interest in resting BMDM (which is consistent with NELF-B KO mice lacking any appreciable phenotype in the absence of challenge). For the 6 genes we analyze quantitatively by RT-qPCR, we now include their baselines in WT vs. NELF-B KO BMDM from RNA-seq (new Figure 3—figure supplement 1=2 each; also see datasets at GSE110279). With respect to their regulation, we now provide for clarity the induction levels for each (which are also unaffected by NELF deletion at the time point of the assay) next to their repression by Dex (revised Figure 3). We note, however, that the effects of NELF on the induction of inflammatory transcriptome is a separate line of investigation pursued by many laboratories including ours: these are complex, dynamic and gene-specific and lie outside of the scope of this study. Here, we have chosen the time point of analysis at which our genes of interest are induced equally in both genotypes which enables us to specifically assess the effect of NELF on repression.

Furthermore, we now performed RNA-seq in WT and NELF-B KO BMDM exposed to 1-h LPS+Dex (n=3 each genotype; GSE110279). New Figure 3 shows that a number of LPS-induced Dex-repressed genes are expressed at higher levels (*derepressed*) in NELF-B KO and nearly all of them *belong in the paused class*. This is a striking result given that RNA-seq generates a different kind of data and quantifies all RNA not just nascent unprocessed transcripts as we usually evaluate by RT-qPCR; yet this gene class-specific trend held up to genome-wide analysis. As control, we do not observe such differential expression for LPS-induced Dex-insensitive genes (new Figure 3—figure supplement 1).

5) Further related to the pause-release mechanism, a study by Zhu et al. (Biochemistry 2011) showed that the master regulator of Pol II pause release (the Cdk9/P-TEFb kinase) is a competitive decelerator of GR transactivation activity and does not interfere with the inhibitory activity of NELF. What is the role of CDK9 on NF-κB and GR binding in the repression of inflammatory genes? Along the same line, one key relevant article (PMID 15879558) was not discussed in the context of the discoveries of this manuscript. In that paper, Luecke and Yamamoto published that GR blocks recruitment of the P-TEFb kinase by NF-κB to effect promoter-specific transcriptional repression. Is there any relationship between these previous discoveries and the mechanisms of LPS+Dex stimulation/repression discussed in this manuscript?

Thank you for pointing this out; we did mention the connection to CDK9 in Discussion, but it is a good idea to make it stronger. We are happy to cite the Luecke and Yamamoto (2005) work along with our Gupte et al. (2013) paper. Our main observation is that CDK9 recruitment to LPS-induced genes is greatly inhibited in the presence of Dex, however, this occurs irrespective of the gene class. We include a new Figure 3that evaluates CDK9 occupancy at the same genes whose expression we assess in Figure 3. Given that NELF rather than CDK9 was a critical distinction between paused and non-paused gene regulation, we focused our efforts on the former. That said, we are open to the possibility that multiple mechanisms contribute to NELF retention: failure to be phosphorylated by CDK9 is a valid mechanism as well as physical interactions between NELF and other components of the GR repression complex. We added requested information to revised Results and Discussion.

6) An old model for nuclear receptor-dependent repression of inflammatory response genes posited re-distribution of coactivators upon nuclear receptor activation. Here, most GR sites are not at NF-κB peaks and could potentially compete for the binding of p300 for gene activation. This mechanism would also result in loss of p300 at p65 peaks and be overcome by overexpression of wild type p300 but not mutant p300. Therefore, the experiments presented in Figure 5 do not clearly demonstrate that local tethering of GR is the cause of reduced p300 and histone acetylation at these sites. Examination of p300 and histone acetylation, etc., in a cell engineered to express a DNA binding mutant that cannot bind to specific DNA sites (and thus recruit p300 to these sites) but can still tether to NF-κB sites, as requested in point 3, above, would be needed to address this concern.

Indeed, we are familiar with the heavily debated ‘competition for CBP/p300’ model and the subsequent studies arriving at very different conclusions. We appreciate the suggestion of using GR mutants however, this will create a new, stoichiometrically and biochemically intractable experimental system; in addition, it is unknown whether GR even needs to be DNA-bound to recruit p300 and evaluating the ability of any mutants to interact with p300 would constitute a project way out of scope of this study. Instead, we proposed another solution – as agreed upon by the Editor – which addresses the same concern within the same primary unmanipulated BMDM in a cleaner way. Indeed, if GR activation simply creates a ‘cofactor sink’, then p300 will be broadly evicted from most/all LPS-induced p65 targets. Hence, we analyzed the recruitment of p300 to the p65 binding sites of LPS-induced genes that were resistant to the effects of Dex. We now show that LPS-induced Dex-insensitive genes retain p300 at their p65 binding sites. Their expression (from our original RNA-seq), p65 ChIP-seq read density distribution and p300 ChIP-qPCR data (in the 3 experimental conditions each) are shown as a new Figure 5.